# New relative sea-level (RSL) indications from the Eastern Mediterranean: Middle Bronze Age to the Roman period (~3800–1800 y BP) archaeological constructions at Dor, the Carmel coast, Israel

Assaf Yasur-Landau[1,2]*, Gilad Shtienberg[3], Gil Gambash[1,2,4,5], Giorgio Spada[6], Daniele Melini[7], Ehud Arkin-Shalev[1,2], Anthony Tamberino[3], Jack Reese[3], Thomas E. Levy[3], Dorit Sivan[1,2]

1 Department of Maritime Civilizations, The Leon. H. Charney School of Marine Sciences, University of Haifa, Haifa, Israel, 2 The Leon Recanati Institute for Maritime Studies (RIMS), University of Haifa, Haifa, Israel, 3 Department of Anthropology, Scripps Center for Marine Archaeology, Center for Cyber-Archaeology and Sustainability, University of California, San Diego, CA, United States of America, 4 The Haifa Center for Mediterranean History, University of Haifa, Haifa, Israel, 5 Leverhulme Visiting Professor, The Institute of Classical Studies, University of London, London, United Kingdom, 6 Dipartimento di Fisica e Astronomia (DIFA), Università di Bologna, Bologna, Italy, 7 Instituto Nazionale di Geofisica e Vulcanologia (INGV)— Sezione di Sismologia e Tettonofisica (Roma I), Roma, Italy

* assafyasur@hotmail.com

## Abstract

This article presents new archaeological observations and multidisciplinary research from Dor, Israel to establish a more reliable relative sea level for the Carmel Coast and Southern Levant between the Middle Bronze Age and the Roman period (ca. 3500–1800 y BP). Our record indicates a period of low relative sea level, around -2.5 m below present, from the Middle Bronze Age to the Hellenistic period (ca. 3500–2200 y BP). This was followed by a rapid rise to present levels, starting in the Hellenistic period and concluding during the Roman period (ca. 2200–1800 y BP). These Roman levels agree with other relative sea-level indications from Israel and other tectonically stable areas in the Mediterranean. Several relative sea-level reconstruction models carried out in the current study provide different predictions due to their parameters and do not model the changes observed from field data which points to a non-isostatic origin for the changes. Long-term low stable Iron Age relative sea level can be seen in Dor, where Iron Age harbor structures remain around the same elevation between ca. 3100–2700 y BP. A similar pattern occurs at Atlit, the Iron Age harbor to the north used continuously from ca. 2900 y BP to the beginning of the Hellenistic period (ca. 2200 y BP). An examination of historical and archaeological sources reveals decline and occasional disappearance of Hellenistic sites along the coast of Israel at ca. 2200 y BP (2nd century BCE), as in the case of Yavneh Yam, Ashdod Yam, Straton's Tower, and tel Taninim. In Akko-Ptolemais, the large harbor installations built in the Hellenistic period were never replaced by a substantial Roman harbor. The conclusions of this research are thus

**Data Availability Statement:** All relevant data are within the manuscript and its Supporting Information files.

**Funding:** AYL - the Israel Science Foundation (Grant ID 495/18 The Maritime Interface in the Bronze and Iron Ages). GS - Murray Galinson San Diego – Israel Initiative; the Israel Institute (Washington, D.C.). TEL - The Koret Foundation (Grant ID 19-0295); Marian Scheuer-Sofaer and Abraham Sofaer Foundation; Norma and Reuben Kershaw Family Foundation; Phokion and Liz Anne Potaminos Family Foundation, Ellen Lehman and Charles Kennel - Alan G Lehman and Jane A. Lehman Foundation; Paul and Margaret Meyer. GG - Leverhulme Trust, UK. G.S. is funded by a FFABR (Finanziamento delle Attività Base di Ricerca) grant of MIUR (Ministero dell'Istruzione, dell'Università e della Ricerca). The funders had no role in study design, data collection and analysis, decision to publish, or preparation of the manuscript.

**Competing interests:** The authors have declared that no competing interests exist.

relevant for the sea-level research community and for the historical analyses of the Israeli and South Levantine coastline.

## 1. Introduction

In the Mediterranean ancient archaeological constructions built adjacent to the shoreline are essential proxies used for reconstructing the Holocene relative sea level (RSL) as summarized by Sivan et al. [1] and Dean et al. [2] for the east and by Vacchi et al. [3] (and references therein) for the west Mediterranean. These sea-level markers date back to the early Holocene and include submerged prehistoric settlement constructions [4–6]. From later periods, coastal structures like fishponds [7, 8], harbor structures and quays built in connection to the sea [9], as well as coastal wells [10–12] were found to be useful indicators for assessing sea-level fluctuations since prehistory.

Archaeological indicators can be translated into sea-level information through the use of high-resolution elevation measurements and dating, both with the minimum uncertainties that can be achieved but also an understanding of the original "functional height" [13], also known as indicative meaning [14] which is the original relation of the structure to past mean sea level (MSL). These relationships differ based on the types of archaeological coastal/marine remains. For example, the interpretation of harbor installations depend on the size of the ships using them [9], while fish tanks as sea-level indicators are a subject of continued debate (see e.g. [7, 8]). The local water table is a critical factor for coastal wells used as sea-level indicators [12]. In many cases, archaeological remains produce terrestrial or marine limiting points providing only upper or lower constraints on past sea level [3, 15, 16].

In Israel, the Holocene sea-level reconstructions are especially robust for the last 2000 years, as the amount of data is relatively plentiful, presented by Sivan et al. [11]; Vunsh et al. [12] and Dean et al. [2] with one study [17] concentrating on the last millennium of RSL indicators. The study by Dean et al. [2] re-assessed all previously studied data and their functional heights, dating, and associated uncertainties for the last 2000 years. This re-assessment followed a protocol developed for the International Geoscience Program (IGCP) [14, 18] with an addition of statistical analysis using a Bayesian statistical model to produce the most up to date RSL reconstruction of Israel during the last 2000 years. Earlier studies, based on archaeological indications [1, 4, 5, 10, 19] presented reconstructed curves for the last 10000 years [4, 6, 10, 20]. These however included very limited number of indicators with large uncertainties for the period between 4000–2000 y BP. Thus, for example, for the Iron Age, starting around 3200 y BP, reconstructed sea levels varied between RSL at about -1.5 [4, 10] to -1.0m [5, 20] and around present MSL according to Raban and Galili [19], based on the data from Dor, Carmel coast, while Sivan et al. [1] presented data from Dor with RSL constraints between -0.4. to +0.6m for the period between 3200 y BP to 3000 y BP.

The current paper aims to address this lacuna in the study of RSL during this long timespan for which there is a dearth of information. It consists of significant new field data from archaeological sea-level indications at Dor, located along the Carmel coast, Israel (Fig 1), and is accompanied by re-assessment of previous relevant archaeological data. The study covers many archaeological periods and significant historical processes: the Middle Bronze Age (MB; ca. 1950–1550 BCE); the Late Bronze Age (LB; ca. 1550–1200 BCE); the Iron Age (IA; ca.1200-530 BCE); the Persian period (ca. 530–330 BCE); the Hellenistic period (ca. 330–63 BCE), up to the beginning of the Early Roman period (ca. 63 BCE to 200 CE).

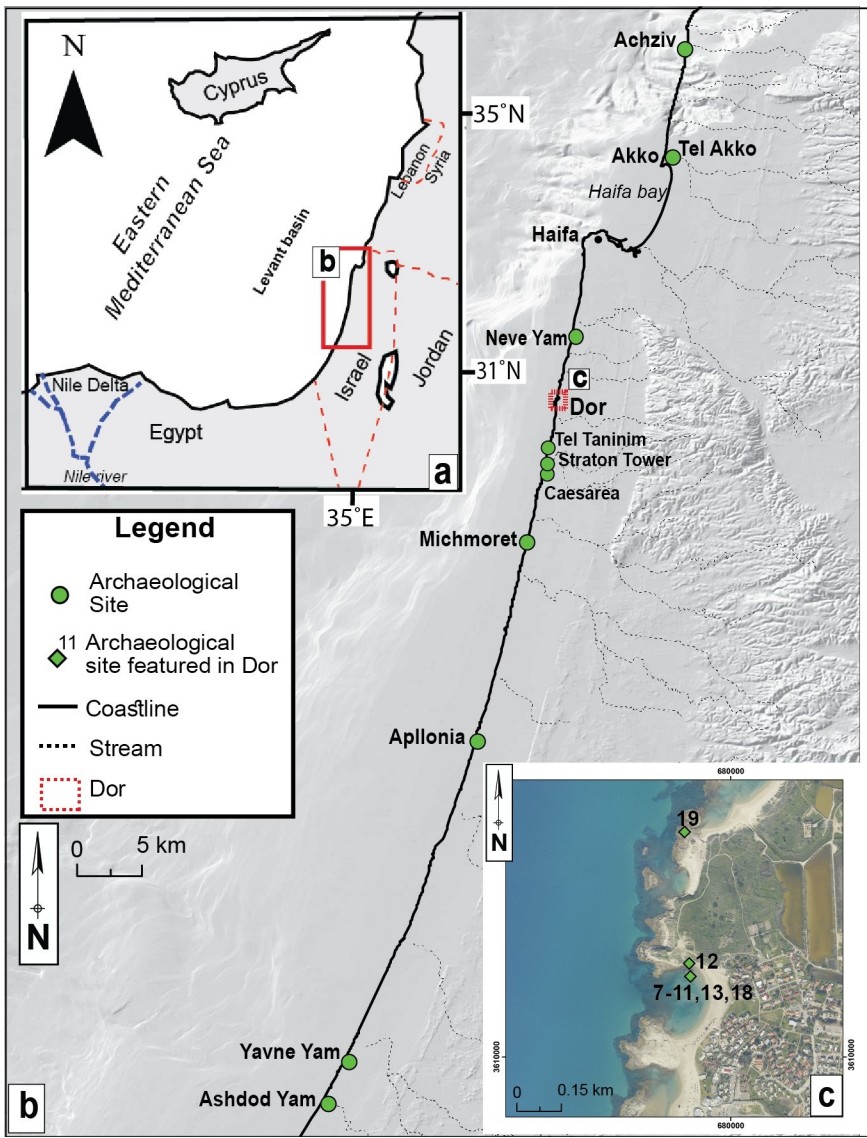

**Fig 1.** Location maps (a) Israel in the SE Mediterranean modified after natural earth (https://www.naturalearthdata.com in the public domain) (b) Shaded relief of the Israeli coast and location of archaeological sea level indicators and locations discussed in the current study (Republished from [21] under a CC BY license, with permission from [the geological Survey of Israel], original copyright [1994]) (c) Aerial photographs of Dor and the location of the archeological structures used in the current study.

## 2. Regional setting: Geology and coastal settlement history from Middle Bronze to Roman times

The coast of Israel in general and the Carmel coast in particular are ideal for reconstructing RSL from the Middle Bronze to the Roman period due to a number of factors: the micro-tidal nature of the Israeli coast (±0.40 m tidal range) according to Davis and Hayes [22] and the Admiralty tide tables [23]; its relative tectonic stability since MIS5e [2, 24, 25], with low present isostatic rates of RSL change, about 0.1 mm/year, assumed consistent for the whole Holocene [1, 2, 17, 26] and the fact that the coast and parts of the submerged zone have been occupied by humans almost continuously during the Holocene [27, 28].

Studies from different disciplines support the notion that the coastal zone of Israel during the period of study have been relatively tectonically stable, including geological mapping [29] along with seismic data [30] and, for the last 130,000 years, from sedimentary and faunal depositions in areas such as the Carmel coast [24] and the Galilee coast [25]. For the last thousands years there is also archaeological [31] and bioconstruction [32, 33] evidence for this stability, which is in contrast to the vertical activity in the coasts of Lebanon in the Holocene. Thus, Morhange et al. [34] notes that "The Rosh Hanikra/Ras Nakoura fault marks the southern boundary of the Levantine vertical displacements, with no evidence for coastal uplift being reported from Israel during the Holocene". Even historical tsunami events were not triggered along the coast but rather in the Dead Sea transform fault in the east or south of Cyprus and Crete in the west [35].

As for Glacial Isostasy Adjustment (GIA) vertical contribution, the models produced for Israel by Lambeck [1, 11, 36] indicates rising RSL throughout the Holocene. Unlike these models, for the last 4000 years, ICE-5G [17, 37, 38] predicts RSL falling to present levels. In all cases, these low rates of RSL change are attributed to GIA, which varies between < 0.2 mm/y for the last 8000 years [1] and 0.15 mm/y in the last 1000 years [17], amounting to ~20 cm RSL rise in 1000 years. For the last 2000 years, the reconstructed RSL that is based on observations indicates small fluctuations above and below present mean sea level as summarized in Dean et al. [2].

The location of the Israel coast, at the very eastern edge of the Mediterranean and a land bridge between Egypt and Southwest Asia, makes the area ideal for connectivity by both land and sea [39]. Adaptation to coastal residence using the coastal aquifer and maritime resources, (and later maritime connectivity and trade) began in the Neolithic period and continued into the Bronze Age. This left a rich record of submerged and coastal settlements from the 8th-2nd millennium BCE, which had first negotiated the challenges of climate changes, rising sea levels, but then were severely impacted by the turmoil of the Bronze Age World System collapse [40]. Coastal habitation recovered during the Iron Age, ca. 1200–530 BCE, with the earliest examples of harbor construction appear in the sites of Dor [40–42] and Atlit [43–45]. The inclusion of the area into the Hellenistic cultural sphere and later the Roman empire further strengthened the coastal sites, with many, such as Akko, Dor, Caesarea and Ashkelon, growing in size and importance [39]. During these periods artificial harbor construction reached an apogee [46], demonstrated in examples like the Hellenistic harbor installations of Akko, Straton's Tower and later Roman Caesarea [39, 47–50].

Dor, the principal site investigated for this study, is located 21 km south of Haifa and 13 km north of Caesarea. It was settled in the Middle and Late Bronze Ages, and remained inhabited also for the Iron Age, the Persian and Hellenistic periods to at least the Late Roman period, some 2300 years of uninterrupted habitation [41, 51–53]. Its rich settlement history as a coastal port site makes it an ideal case for a diachronic study of archaeological proxies for sea-level changes, providing examples of harbor and coastal installations dating from the Early Iron Age to the Roman period. Dor was quick to recover from the collapse of the Late Bronze Age ca. 1200 BCE, trading with Egypt and building coastal fortifications in the (late) 12th century BCE (Iron 1a period) and monumental structures on the tel in the 11th century (Iron 1b period) [52, 54]. The latter were accompanied by contemporary massive coastal fortifications in the south bay of Dor. These fortifications, their foundations now submerged, were previously interpreted as quays. An artificial mole, now obscured by a coat of biogenic rock, served as the main harbor installation of the city [41, 42]. After a hiatus in maritime activities in the 9th and most of the 8th centuries BCE, Dor returned to be a maritime center under the Assyrian empire in the 7th century BCE, it was fortified and a state of the art sea gate was built on top of the earlier sea wall [42]. During the Hellenistic period (3rd-2nd centuries BCE) Dor remained a

strong coastal city, equipped with another new set of fortifications [53]. Underwater excavations at the southern edge of the south bay, as well as geophysical prospections, have uncovered evidence for fortifications guarding the entrance to the bay from the south, likely belonging to this period [55]. The prosperity of Dor continued into the Roman period (1st-3rd Centuries CE), boasting a set of large temples on the tel. The city continued to the North of the tel and included a theatre, large coastal administrative structure, and an industrial zone with pools connected with the sea to the north of the tel [41, 53, 56].

## 3. Methods used for obtaining the new data (Dor)

The current research adds 9 new RSL data points (all those from Dor) to the 13 existing points (Fig 2 and Table 1), thus significantly improving the resolution of sea-level change in the Southern Levant between the Middle Bronze Age and the Roman period. The new RSL points presented below were established by transforming recently excavated archeological constructions elevations and various functions into RSL data. This information was not used so far as sea level indicators.

### 3.1 Underwater archaeological surveying of the new data: Measurements and dating

The Dor underwater structures were excavated after obtaining the proper permits from the Israel Antiquities Authority and the Israel Natural Parks Authority. They were excavated by means of a water dredge system [42]. Measurement and levels were taken using a Leica TS9 Plus Total Station as well as a Leica TS06 Plus with an error of ±5 and ±10 cm respectively relative to Israel Land Survey Datum (ILSD). Subsequently, the ILSD points were converted into local MSL, following Rosen et al. [66] calculations. Based on these finds, that relied on tidal gauges measurements distributed along the Israeli coast between 1958–1984, it was determined that MSL is higher by 8 cm above ILSD due to ongoing sea level rise.

Chronology was obtained by pottery retrieved from the foundation trench of the coastal walls, providing a *terminus post quem* for the construction. The pottery was then compared to the very detailed ceramic sequence of the terrestrial levels at tel Dor, which is also supported by radiocarbon evidence [54, 67] and references therein. Uncertainty is estimated to be no more than ±50 years.

### 3.2 The coastal pool in Dor

A heavy-lift octocopter drone was used to collect photogrammetric data to create a digital surface model (DSM) of the coastal tide of Dor after attaining the proper permits from the Israel antiquities authority and the Israel Natural Parks Authority. Survey data was georeferenced with a South Galaxy G1 RTK-GPS with a vertical and horizontal error no greater than ±3cm creating an orthorectified photomosaic. The pool was a part of a Roman industrial complex excavated by Raban [41]. The pool's period of use was established by analyzing the pottery found in the complex inside one of the plastered basins. While Raban had dated the pottery to the 2nd-early 3rd centuries CE, our examination of this still unpublished pottery assemblage suggested a date within the 2nd century CE for the latest use of the complex [41], or ca. 1,900 ±100 y BP.

### 3.3 Functional height and RSL evaluation

The new archaeological RSL indicators used in this paper are:

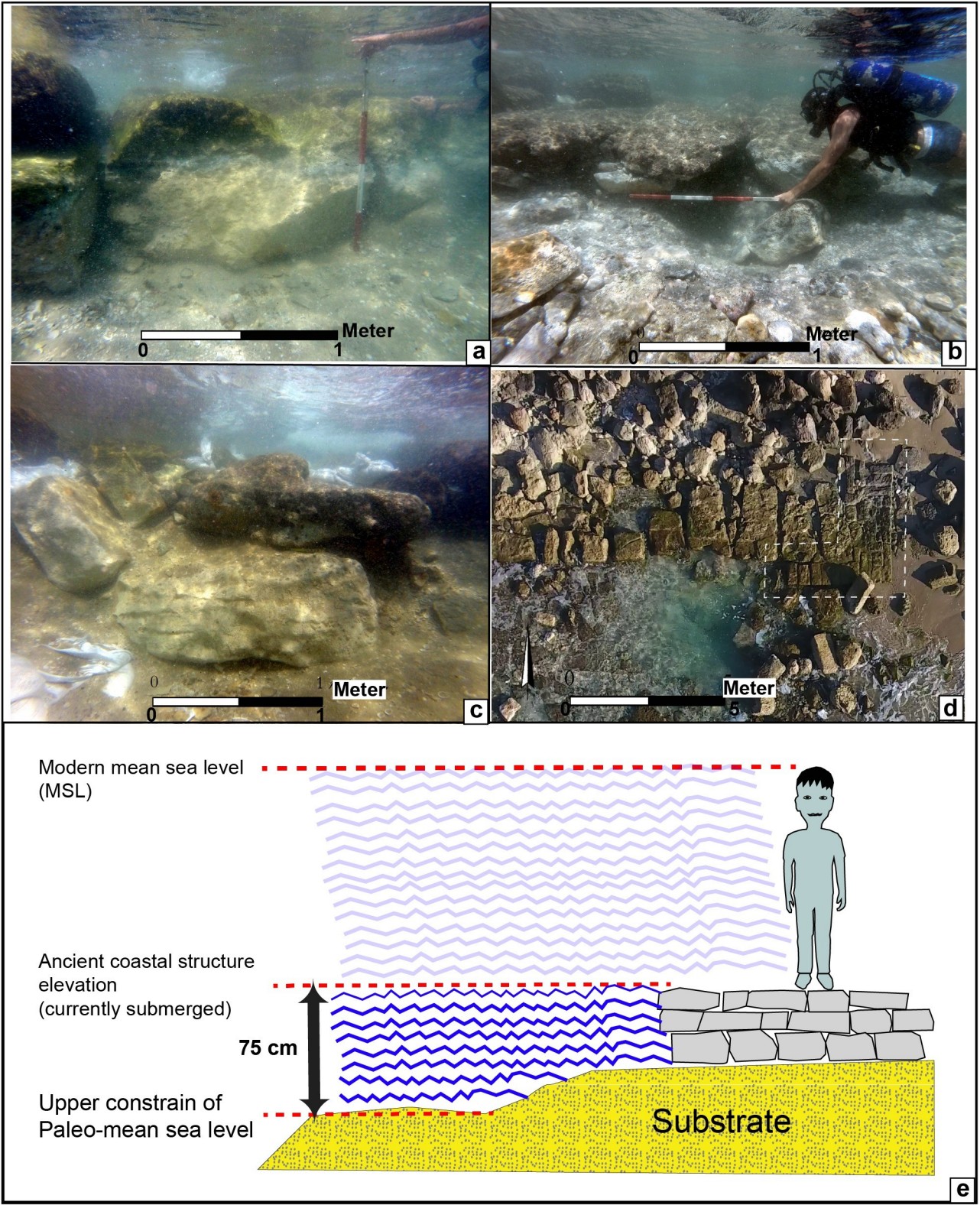

**Fig 2.** Selected examples of the new archaeological constructions used in the current study for establishing RSL; (a) Dor 8—the base of a terrestrial massive fortification wall (W16S-220); (b) Dor 10—base of a terrestrial massive fortification wall (W16S-210); (c) Dor 7—base of a terrestrial wall (W16S-240) (d) Dor 18 –floor level of passage in Assyrian sea gate (W16S-230); (e) an illustration showing the relationship between the coastal archaeological construction and its use for establishing the relative mean sea level.

**Table 1. The Israeli archeological features used in the current study for assessing the relative sea level of the Middle Bronze IIA to Roman period.**

| ID | Site | Site feature and field ID | Archaeological period | Indicator type | Date (BP) | Date uncertainty (y) | Measured elevation (mILSD) | Upper \Lower boundary elevation (mILSD) | RSL assessment value (m) | Total uncertainties (m) | References |
|---|---|---|---|---|---|---|---|---|---|---|---|
| 1 | tel Nami | Coastal Well | MBIIA | Index point | 3,850 | ±100 | -0.7 | na | -0.78 | ±0.40 | Marcus [57] Sivan et al. [1] |
| 2 | Atlit (Fisherman village) | Wall W2 and floor 105 | MBIIA | Upper constraint | 3,850 | ±100 | +1.45 | 0.7 | 0.78 | ±0.33 | Raban [43] |
| 3 | Neve Yam | Byblian anchor concentration | MBIIA | Lower constraint | 3,850 | ±100 | -3 | -3 | -3 | ±0.5 | Galili [58]; Galili and Zvieli [59] |
| 4 | Apollonia | Byblian anchor concentration | MBIIA | Lower constraint | 3,850 | ±100 | -3 | -3 | -2.92 | ±0.32 | Galili [58]; Galili and Zvieli [59] |
| 5 | Atlit north bay | Underwater Tombs | MBII (A-C) | Upper constraint | 3,725 | ±225 | -1 | -1.75 | -1.75 | ±0.5 | Raban [43] |
| 6 | Neve Yam | Cargo of copper ingots and pottery | Iron Ib | Lower constraint | 3,000 | ±50 | -3 | -3 | -3 | ±0.67 | Galili et al. [60]; Yahalom Mack et al. [61]; Arkin-Shalev et al. in press |
| 7 | Dor | Base of terrestrial wall W16S-240 | Iron I a-b | Upper constraint | 3,100 | ±100 | -1.26 | -2 | -1.92 | ±0.32 | Arkin-Shalev et al. [42] |
| 8 | Dor | Base of terrestrial massive fortification wall, Ashlar part W16S-220 | Iron Ib | Upper constraint | 3,000 | ±50 | -0.76 | -1.5 | -1.42 | ±0.32 | Raban [41] (W9); Arkin-Shalev et al. [42] |
| 9 | Dor | Elevation above base of terrestrial massive fortification wall, boulder part 101 | Iron I a-b | Upper constraint | 3,100 | ±100 | -1' | -1.75 | -1.67 | ±0.32 | Raban [41]; Yasur-Landau and Ratzlaff [62] |
| 10 | Dor | Base of ashlar Platform W16S-210 | Iron 1a-b OR Iron IIc | Upper constraint | 2,825 | ±175 | -0.5 | -1.25 | -1.17 | ±0.32 | Arkin-Shalev et al. [42] |
| 11 | Dor | Bottom level of Floor F16K1-111 North of coastal wall W17K1-137 | Iron Ib | Upper constraint | 3,000 | ±50 | 0.4 | -0.35 | -0.27 | ±0.33 | Sivan et al. [1]; Shahack-Gross personal communication |
| 12 | Dor | Coastal well | Hellenistic | Coastal Well | 2,250 | ±150 | -0.4 | na | -0.57 | ±0.54 | Arkin-Shalev et al. [42]; Sharon [63]; |
| 13 | Dor | Hellenistic coastal structure or fortification | Hellenistic | Upper constraint | 2,155 | ±5 | +0.05 | -0.70 | -0.62 | ±0.32 | Yasur-Landau and Ratzlaff [62] |
| 14 | Mikhmoret | Coastal well | Persian | Coastal Well | 2,440 | ±110 | -1.4 | na | -1.73 | ±0.62 | Nir and Eldar [64]; Sivan et al. [1]; Dean et al. [2] |
| 15 | Akko | Top of floor of Hellenistic construction | Hellenistic | Upper constraint | 2,175 | ±75 | -1.1 | -1.85 | -1.77 | ±0.32 | Sharvit et al. [48] |

(*Continued*)

**Table 1.** (Continued)

| ID | Site | Site feature and field ID | Archaeological period | Indicator type | Date (BP) | Date uncertainty (y) | Measured elevation (mILSD) | Upper \Lower boundary elevation (mILSD) | RSL assessment value (m) | Total uncertainties (m) | References |
|---|---|---|---|---|---|---|---|---|---|---|---|
| 16 | Yavneh Yam | Coastal well | Hellenistic | Coastal Well | 2,250 | ±150 | -0.7 | na | -0.56 | ±0.49 | Nir and Eldar [64]; Sivan et al. [1] |
| 17 | Caesarea | Bottom level of lower stone course of round tower in area T/1 | Hellenistic | Upper constraint | 2,200 | ±100 | -1.3 | -2.05 | -1.97 | ±0.32 | Raban [65] |
| 18 | Dor | Floor level of the passage in Assyrian sea gate W16S-230 | Iron Ic | Upper constraint | 2,650 | ±50 | +0.05 | -0.70 | -0.62 | ±0.32 | Arkin-Shalev et al. [42] |
| 19 | Dor | Piscine | Roman | Index point | 1,900 | ±100 | +0.176 | na | +0.256 | ±0.32 | Current Study |
| 20 | Caesarea | Coastal well | Roman | Index point | 1,900 | ±50 | +0.47 | na | 0.05 | ±0.33 | Sivan et al. [11]; Dean et al. [2] |
| 21 | Caesarea | Coastal well | Roman | Index point | 1,900 | ±50 | +0.42 | na | 0 | ±0.33 | Sivan et al. [11]; Dean et al. [2] |
| 22 | Caesarea | Coastal well | Roman | Index point | 1,900 | ±50 | +0.42 | na | 0 | ±0.33 | Sivan et al. [11]; Dean et al. [2] |

The table includes: Site name, archeological features, functional period and uncertainty, the measured elevation of the base, feature boundary elevation for calculating past sea level and total uncertainties: The total elevation uncertainties were calculated based on Dean et al. [2]. For the full details presented in the table, please refer to S1 Table.

a. The base level of archaeological structures such as foundations of walls, fortifications and floor surfaces currently underwater or at the waterfront, were used as an upper sea-level limit based on the assumption that they were originally built above sea level (Fig 2). Considering this assumption, the tidal amplitude for the Israeli coast (±0.3 m; [23]), and winter storm surges, a total of 0.75 m were subtracted from the base elevation (BE) of these upper limit indicators. The sea-level indicative meaning can be described as the following expression: RSL< (BE– 0.75 m).

b. Two assumptions serve in the background of employing the rock-cut pool as a tool for the reconstruction of past RSL: a. when the pool was in service, it would have received seawater through wave activity higher than 0.40 m (which is common for the coast of Israel); and b. the RSL could not have been higher than the elevation of the pool's rim, or lower than its base [7, 8, 68]. Thus, in Dor, the channel feeding the pool with seawater would have functioned between these limits. Elevations of the channel's base rise gradually from +0.1m at the channel's connection with the sea, to +0.75m at the channel's connection with the pool—probably in order to reduce wave-energy and encourage the settling of sand prior to the water's entry to the pool (Fig 3). Therefore, we relied on 30 elevation points measured along the axial center of the first 15 meters of the seaward part of the channel. An average was then calculated for these elevation points, which enabled us to reduce the elevation influence of recent bio-rocks formed on the host sandstone aeolianite platform and other post-usage influences.

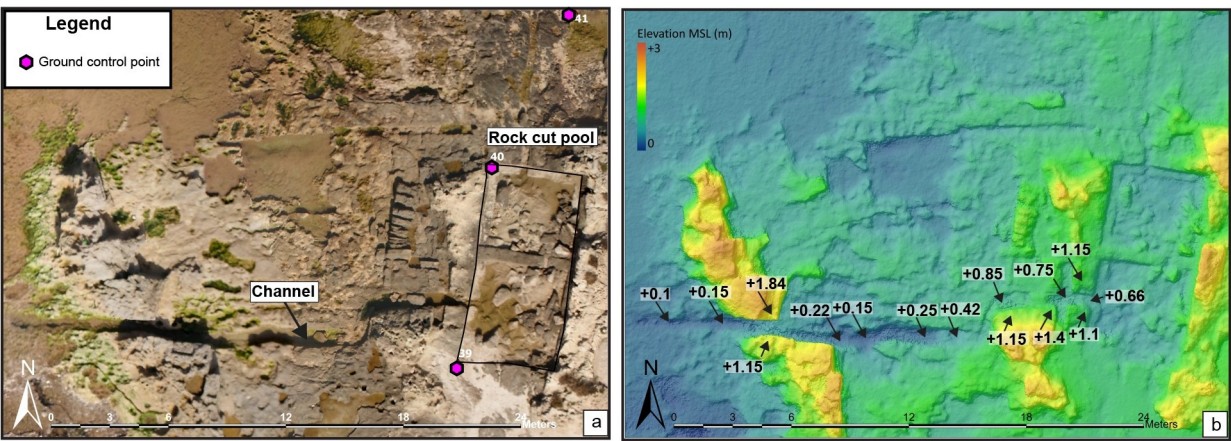

**Fig 3. Elevation model of the Roman rock cut pool from Dor.** (a) aerial photograph of the rock cut pool with ground control points used for georectification (b) a digital elevation model of the pool. The color indicates elevation variation while the black arrow shows elevation changes (relevant to MSL) in and adjacent to the Roman rock cut pool's channel.

### 3.4 Vertical uncertainty assessment

The existing and new data use in the current study has vertical uncertainties that were evaluated based on a variety of factors following Dean et al. [2]. These include: measurement uncertainties, benchmark uncertainty, mean tide fluctuations, and in coastal wells, jar size uncertainty and the assumed past distance from the coastline.

Total uncertainty (2σ) for each indicator ($U_i$) was estimated from the root of the sum of the squares of each uncertainty factor, using the expression:

$$U_i = \left( u_1^2 + u_2^2 + \ldots\ldots + u_{1n}^2 \right)^{1/2}$$

where $u_1 \ldots u_n$ are individual sources of uncertainties for the archaeological remain.

### 3.5 Model predicted RSL for the coast of Central Israel (Dor)

In this work, we model the RSL variations along the coasts of Israel using an improved version of the open source Sea-Level Equation (SLE) solver SELEN4 of Spada and Melini [69]. In SELEN4, the SLE is solved adopting a global, topographically and gravitationally self-consistent pseudo-spectral approach, taking rotational effects into account, and allowing for the migration of the shorelines. A spherically symmetric Earth structure has been assumed, characterized by an incompressible linear Maxwell rheology in the mantle, a perfectly elastic lithospheric layer, and a fluid inviscid core. For a non-rotating Earth, SELEN4 has been successfully tested against independently developed SLE solvers in Martinec *et al.* [70]. The spatial resolution of the simulations performed in this study is in the range of ~ 100–300 km, corresponding to a maximum harmonic degree varying between 128 and 512. These resolutions are expected to adequately describe the long-wavelength spatial variability of the GIA signal in the Mediterranean basin.

In order to test the sensitivity of the RSL predictions in Israel to the history of deglaciation, to the rheological layering of the mantle and to various modeling assumptions, a few different models historically developed within the GIA community have been implemented into SELEN for this study. These include some of the "ICE-X" models developed by WR Peltier in Toronto, *i.e.*, ICE-5G(VM2a) of Peltier [37], ICE-6G(VM5a) of Peltier et al. [71], and ICE-7G_NA (VM7) of Roy and Peltier [72]. In addition, we have employed one of the GIA models

progressively developed at the Australian National University by Kurt Lambeck and collaborators, hereinafter referred to as ANU (the specific ice history adopted here was kindly provided to GS by Anthony Purcell in November 2016). All these models are implemented into SELEN4, so that the outputs obtained may differ slightly from the results published by other authors, due to differences in the resolutions adopted, in the assumptions about the rheological layering and its discretization, and in the rotational theories adopted (see [69]).

## 4. Results

This study produced new relative sea-level indicators and presents them with previously studied indicators which in some cases were also re-assessed. The entire dataset analyzed includes upper and lower limits of man-made structures that originally were built above sea level, coastal pools, coastal wells, and concentrations of cargo or anchors (Table 1). Each of these types of data requires different interpretation to be useful RSL data, so these calculations took into consideration the functional height of each indicator (see: [1, 2, 11, 12]).

### 4.1 New data points

The new data presented here from the environs of tel Dor area research product of the underwater excavations and coastal survey directed by Yasur-Landau (in 2016–17) as part of the Tel Dor joint expedition and by Yasur-Landau and Levy (in 2018), a part of the Tel Dor underwater excavations project of the University of California San Diego and the University of Haifa Koret collaboration. The excavations, both the underwater and the land survey, provide new data mainly on archaeological indicators for Iron age sea levels, a period from which very little data existed before e.g. [1], and new RSL reconstructions of the early Roman period.

Dor 7 is the base of terrestrial wall W16S-240, made of limestone blocks. It is dated by pottery found in its foundation trench to the Iron I a-b period, i.e. to the late 12th to the 11th centuries BCE. Stratigraphically it is earlier then the terrestrial massive fortification wall, W16S-220 of Dor 8 [42]. Base elevation of terrestrial wall relative to ILSD: -1.26 m.

Dor 8 is the base of the ashlar stone construction of the coastal massive fortification wall, W16S-220. This part is currently submerged underwater. This wall is mostly built on coastal sand, while the western part of it is built on top of sandstone (locally named kurkar). The fact that the courses of this wall are leveled, despite resting on different bedding, indicates that no subsidence occurred. It is dated by Iron Ib (i.e. 11th-early 10th centuries BCE) pottery found in its foundation trench in an underwater excavation (wall W9 in [41, 42]). Base elevation of ashlar stone construction relevant to ILSD: -0.76 m.

Dor 9 is the base of the same fortification wall, yet a part extending further to the east and built of large boulders. It is currently located on land, yet its base is below water level. As it is a direct continuity of wall W16S-220 (Dor 8), it is also dated to the Iron 1b period, i.e. 11th-early 10th centuries BCE [62]. Base elevation of fortification wall relevant to ILSD: -1 m

Dor 10 is the lower elevation of ashlar paving blocks connecting the coastal fortification wall with the Iron Age mole (Platform W16S-210). This is a terrestrial feature, with blocks 30cm thick resting on sand. Their date may be either from the original construction of the sea wall, Iron 1b (ca. 11th-early 10th centuries BCE), or from the time of the construction of the sea gate, during the Iron IIC period (7th century BCE) [42]. Elevation of ashlar paving relevant to ILSD: -0.5 m.

Dor 11 is the Bottom level of terrestrial Floor F16K1-111. The floor is located in the inner side of the coastal fortification (north of coastal wall W17K1-137, designating the inner face of the coastal fortification wall W16S-220 (Shahack-Gross personal communication). The pottery on the floor belongs to the Iron 1b period. Elevation of floor relevant to ILSD: +0.4 m

Dor 12 is the bottom of a well. Initially dated to the Iron Age, a reevaluation of the construction methods of this well, and especially the stones used, show that they are dissimilar to Iron Age construction technique, yet show clear affinities of Hellenistic masonry [41]; (see also [63]). Base elevation of well relevant to ILSD: +0.4 m.

Dor 13 is the bottom level of a collapse inside a sturdy Hellenistic coastal structure or fortification located on the coast at the base of the tel by the south bay. This collapse presumably represents a destruction level of the second half of the 2nd century BCE, as it contained pottery, coins of the middle to late 2nd century BCE, and arrowheads [62]. Base elevation of structure relevant to ILSD: +0.05 m

Dor 18 is the floor level of the paved passage in the Assyrian-style sea gate W16S-230. Similar paved passages were found on land at Dor and Megiddo, dated to the 7th century BCE, the Iron IIC period [42]. Floor elevation relevant to ILSD: +0.05 m

Dor 19 is a channel leading from the sea to a fishpond (Fig 3), part of a Roman fish processing complex previously identified as a purple dye factory [41]. A reevaluation of the pottery [73] found in the complex indicate that it was active in the 2nd century CE. For evaluating RSL the current study uses the average elevation of 30 measured points along the channel base in the first 15 meters of its seaside. The channel's average elevation relevant to ILSD is +0.176 m ±0.32 m uncertainty.

## 4.2 Summary of new data points

All the new indications presented in the current study (9 out of 22 data points; Figs 4 and 5) are from the area of Dor along the Carmel coast (Fig 1). 7 out of the 9 new points are terrestrial limiting RSL points, meaning structures that were originally built on land with no known connection to the sea, but at present are submerged in the bay south of the tel. Only data point 19 north of tel Dor is an RSL index point since it was originally a channel built to supply sea water to the pools as a part of the Roman industrial complex (Fig 3). It is well dated based on ceramic typology to the Early Roman Period (see in the methods above).

Most of the structures presented in the overall reconstruction, both the new data from Dor and those from other areas are currently at or below MSL, but since the majority are terrestrial limiting data, the current suggested sea-level position for the period between ~ 3800 y BP and ~2200 y BP was above -3.0 m MSL and probably closer to -2.0 m MSL followed by abrupt sea-level rise to near-present levels in the early Roman period, around 2000–1800 y BP (Fig 5).

Among the previously published data our new data is presented for a fuller reconstruction and context, three RSLs were obtained by coastal wells from Caesarea dated to the early Roman period based on calculations by Sivan et al. [11]. Since the hydrological situation in Caesarea is different from the rest of the coast, the RSLs used here are those originally calculated. Four additional coastal well RSLs were re-calculated assuming that they were located up to 200 m from the coastline when functioning (see Table 1). The estimated past RSLs follow the equation suggested in Sivan et al. [11] and the offset values between the modern water table and sea level follow the model of Vunsh et al. [12].

Ancient concentrations of heavy objects (e.g. ingots) from shipwrecks or anchorages that approximate the location of a grounded ship can provide the lowermost possible sea level at the time of grounding [1, 4, 5]. This relies on the assumption suggesting that such heavy artifacts remained in situ after grounding [5, 58]. These remains were used only as lower RSL constraints (see Table 1).

## 4.3 Model predictions results

Fig 6 shows a set of RSL predictions for Dor between 6000–0 y BP computed for this study by the SELEN4 SLE solver assuming different GIA models, as described in section 3.5 above. The

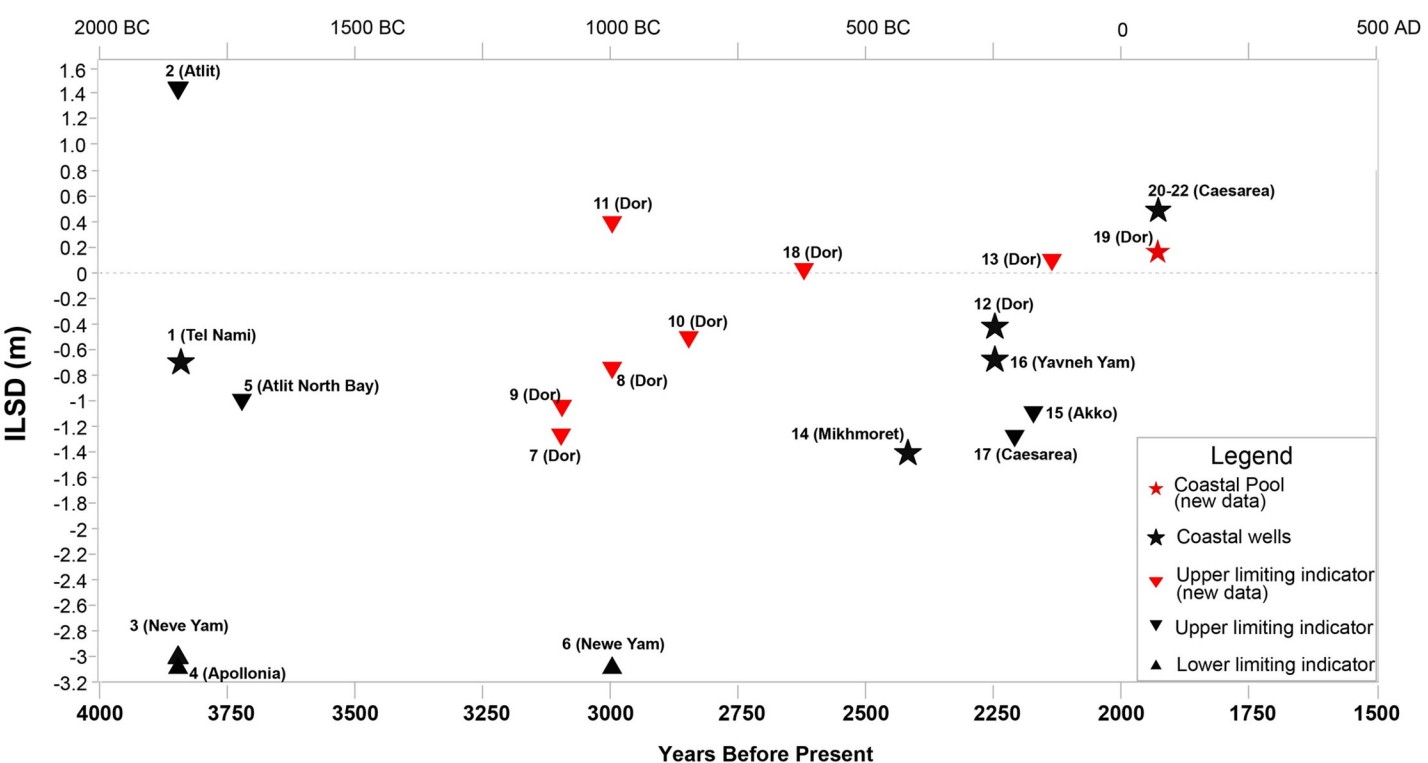

**Fig 4. Base elevation and timeline of the archaeological constructions used in the current research for evaluating relative sea level.**

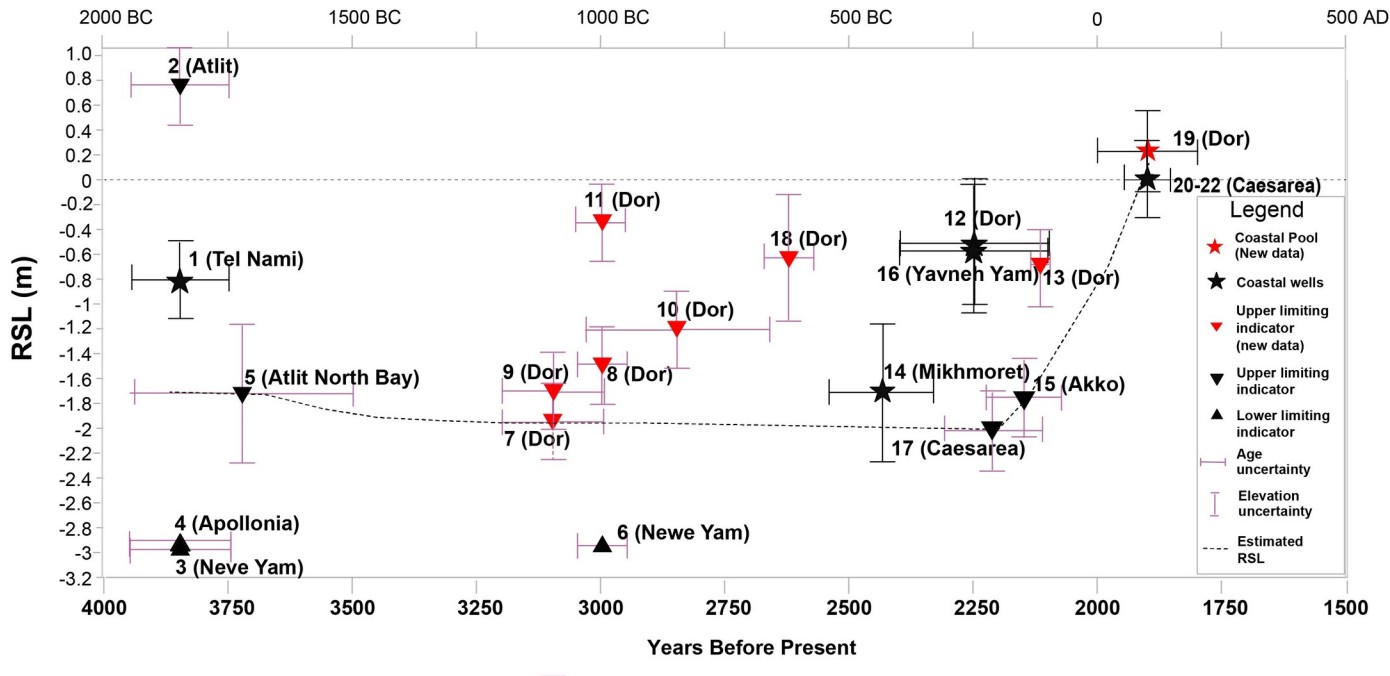

**Fig 5. Computed relative sea level for the coast of Israel with the chronological and vertical uncertainties.**

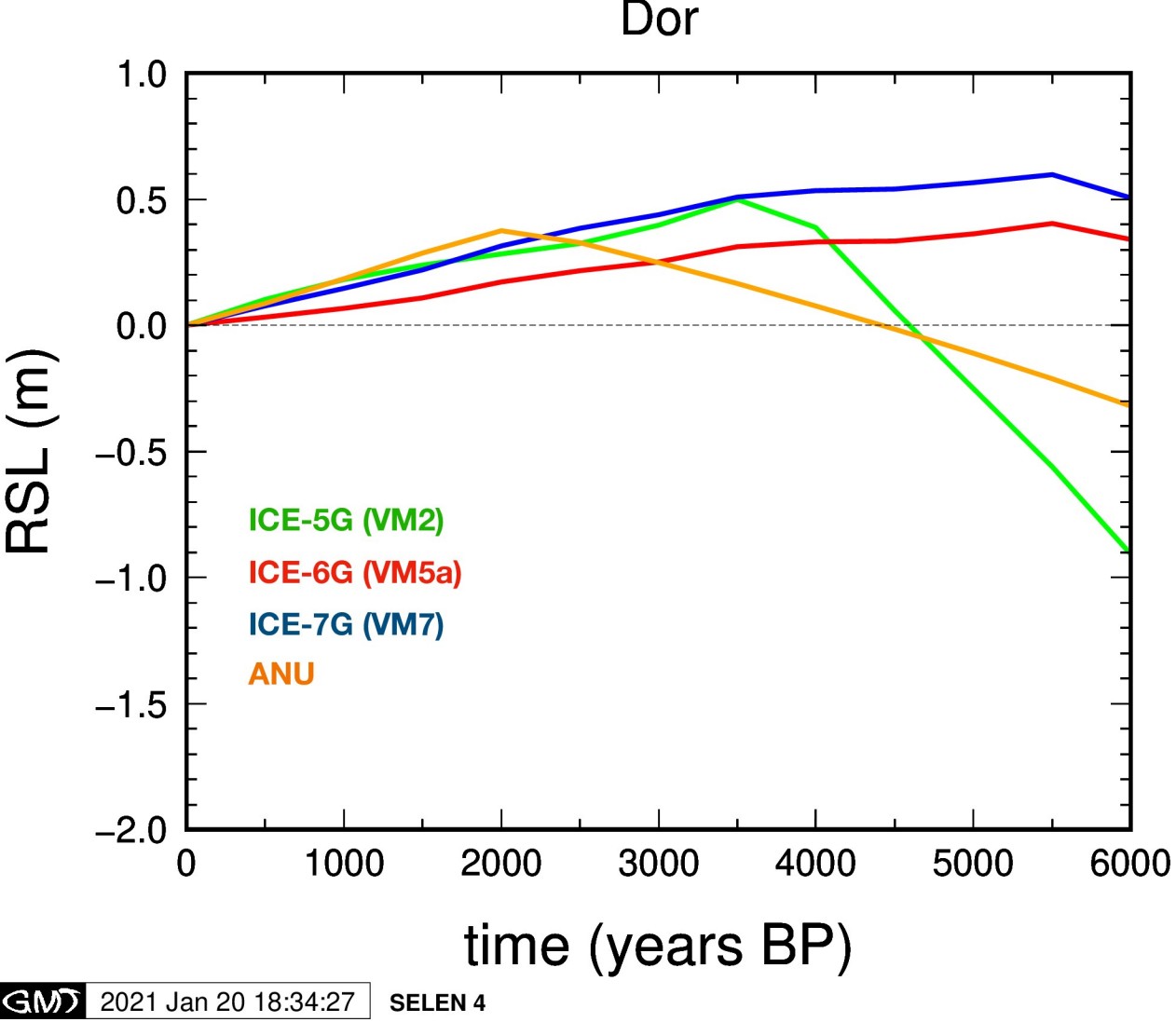

GMD  2021 Jan 20 18:34:27  |  SELEN 4

**Fig 6. Synthetic RSL reconstructions at Dor obtained with the SELEN4 SLE solver.** Models ICE-5G(VM2a), ICE-6G (VM5a), ICE-7G (VM7) and ANU have been considered, as summarized in the main text. All the four GIA models have been implemented into the SELEN4 code.

ICE-X class of models predicts a constant sea-level fall for the past 3500 years, at a rate of about 0.10–0.15 mm/y. The RSL curve obtained with ICE-5G(VM2) shows a sharp highstand of about +0.5m at 3500 y BP, preceded by a fast rise in sea level, while both ICE-6G(VM5a) and ICE-7G(VM7) predict an highstand at about 5500 y BP. Conversely, with ANU we obtain a recent highstand at about 2000 y BP, followed by a sea-level fall at a rate similar to that obtained with the ICE-X models. The differences between these RSL model reconstructions are due to several factors, including different deglaciation histories of the late-Pleistocene ice complexes and different assumptions for the Earth viscosity profile or lithospheric thickness. Therefore, the spread between the RSL curves in Fig 6 can be viewed as a measure of overall uncertainty for GIA modeling (see [74]) in this region.

In Fig 7 we compare RSL observations at the site of Dor with synthetic RSL curves computed with the ICE-6G(VM5a) and ICE-7G(VM7) GIA models. In both cases, we compare the RSL curve computed by the SELEN4 SLE solver with an independent solution using the

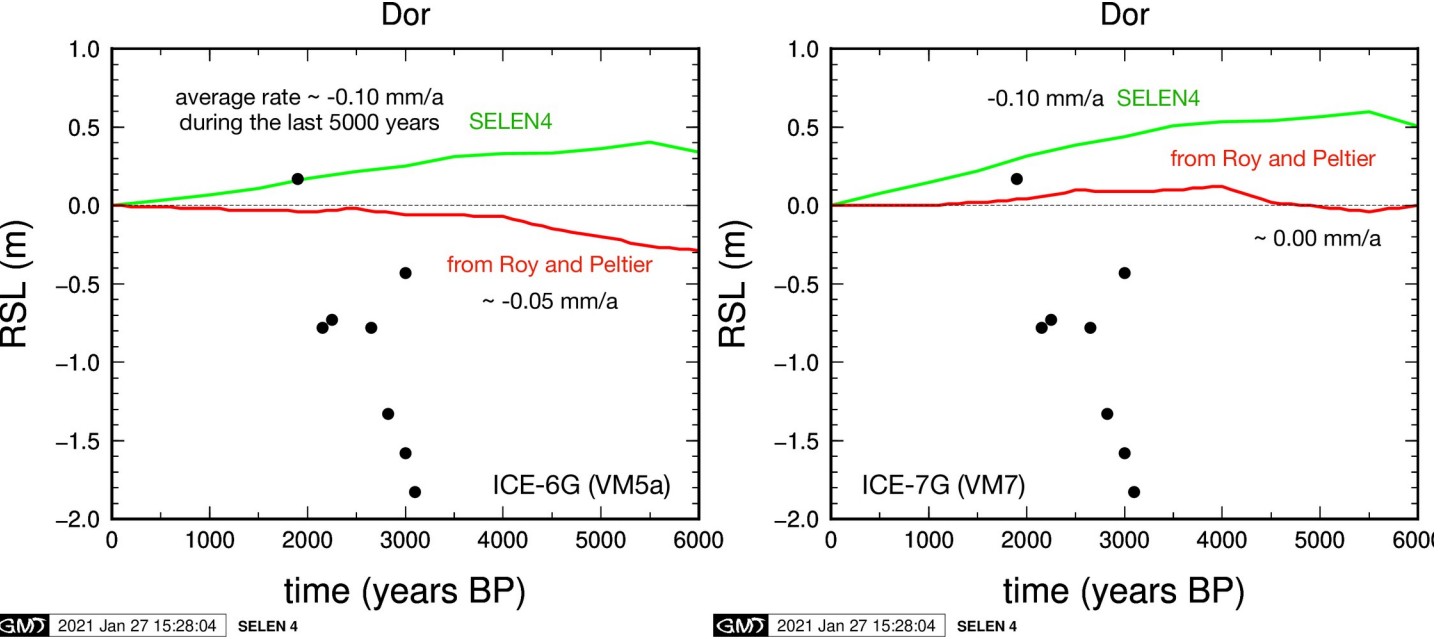

**Fig 7. Synthetic RSL reconstructions at the site of Dor according to the ICE-6G (left) and ICE-7G (right) GIA models.** Each panel shows results obtained with the SELEN4 program by GS (in green) and those using the parameters used in Roy and Peltier [38] red, which were provided by WR Peltier for the current study. In both cases, the models are not in agreement with RSL observations at the Dor site, represented by black circles (see Table 1).

technique from Roy and Peltier [38] provided to GS by WR Peltier for the current study. The discrepancy between RSL predictions for these two independent solutions of the SLE is of the order of 0.1 mm/y vertical change over a time range of 6000 y. This is likely due to different numerical solution schemes and discretization approaches; however, according to Melini and Spada [74], such a level of disagreement between SLE solutions lies well within the expected range of GIA model uncertainties. While the RSL index point at 1900 y BP (see Table 1) is in good agreement with the range of GIA models, all RSL observations between 2000 y and 3000 y do not agree with the GIA predictions, showing a rapid sea-level rise at a rate of over 1 mm/y that is not reproduced by any of the considered models.

## 5. Discussion

### 5.1 RSL of Israel from the MBA to the Roman period (ca. 3800 to 1800 y BP)

Unlike the RSL reconstructed for Israel over the last 2000 years in other research [11, 12] and for the last 1000 years [17], previous studies produced very few RSL indications [1] from between 4000 y BP to 2000 y BP (the MB and IA periods to Early Roman period). Those that were published e.g. [4, 10, 20] have relatively large uncertainties due to limitations on measurements and dating methods.

The current paper better constrains RSL in Israel from the period of 3800 y BP to 1800 y BP by introducing significant, new archaeological relative sea-level indicators, presented in context with re-assessed existing data. All the new data points are from Dor, Carmel coast (Fig 1). Apart from the pools north of the tel (Fig 3) where the channels bringing sea water are considered RSL index points, the new data are terrestrial limiting points [75]. Other points included in the reconstruction were previously published and apart from the three wells in Caesarea [11] they were re-assessed mainly for chronology in the current study (Table 1). The terrestrial limiting points (Fig 5) are an upper limit on sea level, which therefore must have been lower.

Both upper and lower limits from this study indicate that RSL from ~3800 y BP up to ~2200 y BP was between -2 to -3 m MSL. These low levels existed up to ~2200 BP (Hellenistic period), then RSL began to rise. This sea level change is best confirmed by RSL difference between the recently published Hellenistic harbor installations [48] in Akko, dated to 2175 BP, and the pools of Dor, dated to the Roman period. While the Akko port installations are at an elevation of 1.1 m below MSL [see Table 1; Fig 4], the pools of Dor are at near-present RSL [2, 11, 26].

The indication of low sea level around -2.5 m during a period of about 1000–1500 years can be connected to potholes and abrasion notches investigated by Goodman-Chernov et al. [76]. These geomorphological sea-level indicators were identified in few sites along the Israeli coast including Dor, all in depths of ~- 2.5m. These geomorphological features indicate a period of prolonged sea-level stability [77]. They are not dated, but their depth, and the fact they need a relatively long period (depending on the of the rock characteristics) of stable sea level to form, can suggest their age when considered with our new sea-level indicators from Dor.

This paper introduces a high resolution (in elevation and the dating) index point from the Roman pool at Dor. The pool is located close to present MSL, north of submerged structures dated variously from Middle Bronze up to the Hellenistic period. Unlike the Roman rock-cut pools that were thoroughly investigated [7, 8, 68] in the Tyrrhenian Sea, the pool in Dor (Fig 3) is part of industrial complex and consists of a basic construction fed by a channel that gradually rises from +0.15 m in connection to the sea, up to +0.75 m where it connects to the pool. RSL was calculated in this case by using an average of 30 total station points, measured along 15 meters of the channel base where it connects to the sea. This location is significant for evaluating past sea level as a location free of archeological debris, sediment, and biogenic rocks, at a site with many indicators from other time periods. The uncertainty of the index point includes the range between the means of the highest to lowest tide (mean high water spring to mean low water spring tide) for the Israeli coast, evaluated to ± 0.30m [23] as well as uncertainties linked with the elevation measurement [2], and evaluated to be ± 0.32m.

The proximity of the other, submerged indicators from older periods with the Roman pool strengthens the reliability of the reconstruction and shows that vertical tectonic movement postdating the Roman period is not responsible for the RSL fluctuations detected. If vertical movements were responsible then we would expect to see indicators from both periods submerged at the same level. Likewise, no corroborating evidence exists elsewhere on the coast for abrupt movement occurring between the two periods (see section 2 above). The abrupt seal-level rise which started during the Hellenistic period along the coast of Israel came to end at a certain point during the Roman period, and since that point, for roughly 2,000 years, the RSL has remained the same, with the exception of several short-term fluctuations, as shown by Toker et al. [17] and Dean et al. [2]. These new finds and past records presented here cannot be explained so far by a known mechanism.

## 5.2 RSL observations vs. predicted models

There is a significant discrepancy between our observed RSL data and the new iterations of the computed models which we present together in this paper: Low observed levels in the studied period relative to the models. Vertical tectonic activity is not an ideal explanation for this discrepancy, since the coast of Israel is considered relatively stable based on geological conclusions [29] and seismic data [30] together with sedimentological indicators from the Last Interglacial [24, 78] and archaeological indications in the last thousand years ([2] and reference therein).

To briefly summarize the results of previous studies, for the east Mediterranean and especially the coast of Israel there have been various predicted model reconstructions: in Sivan et al. [1, 11] Lambeck computed Holocene RSL between -0.2 m to -0.4 m ca. 2000 y BP, -0.6 m to -0.8

m ca. 3000 y BP and around -1.4m ca. 4000 y BP as presented by Sivan et al. [32]. Although the trend there shows rising sea, a weak highstand of approximately 0.20 m has been suggested. The modeled results of Lambeck and Purcell [36] instead show no highstand in Israel, and the observed results [2] for the last 2000 years are in overall agreement with these models. However, in Toker at al. [17] the ICE-5G(VM2a) of Peltier [37], computed using the SELEN program [79] presents a RSL highstand above present MSL at ca. 4000 y BP and RSL falling since then.

For the current study, four curves have been computed for the coast of Dor by G. Spada and D. Melini, based on the ICE-X models of Peltier and co-authors and on the ANU model of Lambeck and collaborators (Fig 6), whose features are discussed in Section 4.3 above. All the four are in disagreement with the current observations from Dor indicating low RSL (about -2.5m) between around 3500 BP to 2200 BP, and relatively abrupt RSL rise from around 2200 BP to 1800 BP to near-present RSL. In order to investigate the discrepancy between the model and the observations, two more curves were computed with the SELEN code for the coast of Dor (Fig 7), based on the ICE-6G and the ICE-7G models, and have been compared with corresponding curves provided by WR Peltier, using the technique of Roy and Peltier [38]. None of them fit the observations.

The fact that even the same models, by using different parameters, realizations and computer implementations (such as the use of SELEN4 vs. a different SLE solver), can yield different RSL reconstructions, is well presented by the model computations of Roman period RSL in Italy compared to interpretations of fish tanks [68]: ICE-5G and 7G predict similar elevations of -0.50m and -0.48m while ICE-6G predicts -0.20 m. The ICE-6G reconstruction (Fig 7) derived from the method of Roy and Peltier [38] for the coast of Israel is the closest to the Israeli observations but still the predicted RSLs are higher by 2.0 to 2.5 m in the time period the current study focuses on. Along coastlines located in the far field (distal from the ice sheets) the GIA signal is generally small, but even a small difference in the model details such as spatial resolution, mantle layering, etc. can lead to different patterns.

In the rest of the Mediterranean, and mainly in the Western Mediterranean, Holocene RSL predictions based on ICE-5G are rising towards present MSL with no highstand [3]. Only in the Gulf of Gabes, south Tunisia, the observations indicate a highstand of about 1.7 ± 0.3 m to 1.4 ± 0.4 m around 5500 y BP in agreement with the ICE-5G based on a three-layer approximation of the multi-layered viscosity profile VM2. Comparison of the Vacchi et al. [3] data set with predictions of the ICE-6G (VM5a) and ICE-7G (VM7) in the same location was carried out by Roy and Peltier [38], and predicted no highstand with the ICE-6G and a decimetric highstand when using the ICE-7G. In Fig 4 of Roy and Peltier [38] showing RSL change since 2000 y BP, ICE-6G (VM5a) or by ICE-7G (VM7), indicate falling sea levels of 0.25m along the Israeli coast. All Holocene observations along the coast of Israel, apart for ~30 cm RSL oscillation above MSL in the Byzantine period [2], and almost all RSL indicators in the western Mediterranean, show a rise towards present level with no indication of highstands [3].

In general, models are not able to predict centennial fluctuations so some of the sea levels found in the continuous Israeli data for the last 2000 years [2], and presented in the current paper for earlier periods, may not appear in the models for this reason. Various mechanisms to explain fluctuations have been suggested, e.g. [2, 17] for the last 2000 years, but so far with no means of verification. The inability, however, of the models to detect the longer-term low sea levels this study presents for the pre-Roman time period requires further investigation.

## 5.3 The Israeli RSL in Mediterranean perspective

Mediterranean analogues for the sea-level observations relating to the time period the current paper deals with have to be taken from areas considered relatively tectonically stable with low

GIA rates of vertical change per year. Such areas are not common, since large parts of the Mediterranean are tectonically active. Anzidei et al. [78] mapped all vertical movement in the Mediterranean and constrained the areas available for comparison.

In the west Mediterranean, the coasts of south France and north Corsica are among those deemed relatively stable. Along the French coast, sea-level observations between 4000 y BP and 3000 y BP presented by Lambeck and Bard [80] indicate levels varying between -2.0 m to about -1.0 m with large uncertainties, rising to present levels around 2000 years ago (Roman period) which are, in the frame of the uncertainty, not too far from the Israeli results. Morhange et al. [81] concluded slightly different RSL rise based on his study in Marseille harbor. In this study the authors present a rise in RSL from about -1.5m around 4000 y BP up to ~ -0.7m ca. 1500 y BP (Late Roman). From Fréjus, southern France [82], the Roman period RSL data are in agreement with the Israeli levels; from the first and the second centuries CE the levels in Fréjus vary between -40 ±10 cm to -26 ±10 cm, but for earlier periods there is no agreement, since based on previous studies Morhange et al. [82] present very slow RSL rise from around -0.8 m in the Hellenistic period (ca. 2400 y BP) to about -0.2 cm in the Roman period ca. 1700 y BP. Vacchi et al. [3] present for the coast of southern France the lower levels of -1.5 m ±0.4m at 4000 y BP rising to -0.8m ±0.4m at 2000 y BP which are closer to the Israeli data. In north Corsica observations point to sea level lower than -1.0 m ca. 3700 y BP rising slowly and still were lower by about -0.5 m 2000 years ago. For the whole of Corsica, Vacchi et al. [83] concluded RSL of -1.1 ±0.3 m at 3500 y BP rising slowly to -0.8m ±0.3 m at 2100 y BP. In both studies mentioned, the RSLs of Corsica are slightly higher in the relevant period (4000 to 2000 y BP) relative to those obtained in Israel but both records have large uncertainties. Apart from the data from Fréjus [82] in south France and in Corsica, the data is generally in agreement with RSL between -1.5m and -1.0 m ca. 3500 y BP which is shallower than the levels between -3.0 m and -2.0m obtained in Israel. The west Mediterranean data do not present a "jump" ca. 2200 y BP but unlike the Israeli data, that most of it is from the same site of Dor, and the rest from close areas, the data both in France and in Corsica is from various, relatively remote sites.

Unfortunately, the nearby coasts of Lebanon and Syria cannot be used for comparison since north of Rosh Hanikra, Israel, the coasts are tectonically active and those indications dated from 2700 y BP to 1400 y BP (the 6th century CE) are elevated to +0.80 m - ±0.40 m [32, 34]. Therefore, the elevations presented in harbors like Tyre and Sidon for these periods cannot be used for comparison with Israeli sea levels.

## 5.4 The historical impact of RSL changes on the coast of Israel

The abrupt RSL rise from around -2 m (relevant to MSL; Fig 5), seen at Akko, to close to present mean sea level as seen in Dor (Fig 5), in the course of 200 to 400 years between the Hellenistic and Roman periods, raises questions on both the influence of this rise on coastal structures, mainly harbors, and also horizontal coastline changes.

The first question, regarding the influence of the abrupt sea-level rise on harbors and coastal structures can be studied through historical evidence and archaeological remains. The Dor data shows a stratification of sea-related structures along the coast. The earliest wall, now submerged, belongs to the late 12th or early 11th BCE century. It was built over by massive, 11th century BCE coastal fortifications, the lower part of which is now submerged, too. These fortifications were still in use in the 7th century BCE when a new sea gate was built on top of their stone foundation. All these structures had similar elevations indicating long-term RSL remained stable. Similar longevity of Iron Age harbor facilities can be seen in the Iron Age port of Atlit (Fig 1), constructed in the 9th century BCE [44, 45], yet maritime activity within the artificial harbor basin, indicated by the deposition of pottery, continues at least until the 4th

century BCE [43]. However, no later structures dated to the late Hellenistic or Roman period are associated with this harbor. In both cities (Atlit and Dor) no Roman port was built, and there are no Roman fortifications.

The more direct impact of the rapid rise of RSL would have been felt in the sphere of marine installations, and the economy which closely relied on them. Settlements which employed built coastal features would have been vulnerable as far as their maritime related endeavors were concerned, since the installations on which they relied would have gone out of use at a rapid pace, even if they attempted to repair and adjust them to the rising sea level. Both the means to protect harboring ships, such as the sea-walls of Akko-Ptolemais [48], and the installations devised to facilitate the loading and unloading of cargo, such as the quays of Dor and Straton's Tower, would have been rendered dysfunctional as a result of the significant rise in RSL. Larger cities with more varied economies, such as Akko-Ptolemais and Dor, would have been more resilient, and continued to exist during the challenging period, though their maritime-based activity would have been reduced. But a long list of smaller coastal settlements show clear signs of collapse, including Yavneh Yam, Ashdod Yam, Straton's Tower, and tel Taninim.

The decline and occasional disappearance of multiple Hellenistic sites along the coast of Israel, witnessed during the 2nd century BCE, has not been heeded enough by modern research. It should be highlighted, and attributed, at least in part, also to effects caused by the rising sea level. The port site of Yavneh Yam (Fig 1) shows manifest decrease in the settlement activity in the acropolis between the Hellenistic and Early Roman periods, including the end of use of the Hellenistic well, while agricultural activity north of the site continued throughout [84, 85]. In Ashdod Yam (Fig 1) the monumental Hellenistic structures on top of the Acropolis, no doubt a fortified complex related to control over the anchorage, were destroyed at the end of the 2nd century BCE, and were not built again in the early Roman period [86]. Straton's Tower (Fig 1), which had probably contained elaborate harbor facilities in the early Hellenistic period, was all but deserted by the 1st century BCE [87]. The mammoth Roman harbour of Caesarea (Fig 1), built at the end of the 1st century BCE by Herod, was located to the South of Straton's Tower, along a hitherto undeveloped coast. Also at tel Taninim (Krokodeilon Polis), a town and an anchorage located between Dor and Caesarea, there is a long gap of occupation between the late Hellenistic (1st century BCE) to the early Byzantine period (ca. 300 CE) [88]. A gap in Akko-Ptolemais exists in the large, well designed harbor [47, 48], built earlier in the Hellenistic period, which possibly saw some later repairs to its breakwater but was never replaced by any substantial Roman harbor works, as seen by the Roman and later pottery found in it [89, 90]. Coastal fortifications, such as the ones existing in Dor, would have also been diminished by rapid sea-level rise and that may well be the reason for the lack of fortifications in Dor during the Roman period.

The dire situation of the coastal settlements during the 1st century BCE, without safe harbors available for mooring and anchorages exposed to the hydrodynamics of the sea, is vividly depicted in the writing of Flavius Josephus, showing the situation before Herod the Great began the construction of Caesarea in 22BCE: "This city [Straton's Tower] is situated in Phoenicia, on the sea-route to Egypt, between Joppa and Dora, which are lesser maritime cities, and not fit for havens, on account of the impetuous south-western winds that beat upon them, which rolling the sands that come from the sea against the shores, do not permit smooth landing; but the merchants are generally forced to ride unsteadily at their anchors in the sea itself [91]" (see also [92]).

The consequences of infrastructure degradation due to sea-level change would have affected most immediately the harbor city's routine commercial activity, and, therefore, its economy. To be sure, political implications would have followed, and the coastal cities of the Southern Levant may well have found themselves in a weakened position when they had to face the

expansionist ambitions of the Hasmonean dynasty, and later the encroachment of the Roman Empire. On a more practical level, the entire coastal Southern Levant appears to have put on hold all prospects for an improved maritime interface. With the relics of earlier installations still protruding dysfunctional out of the water, knowledge of the rising sea level and its rapid pace would have been widely available, serving as a painful reminder of past glory for such cities as Dor and Akko, and as a clear discouragement of investment in coastal facilities for the entire area. This may well explain the fact that no considerable harbor logistics were developed after the building of the seawalls and quays in Akko-Ptolemais [48], probably during the late 3[rd] or early 2[nd] century BCE. It may also justify the solidity of the Herodian harbor structures in Caesarea, often considered excessive by modern research [93].

## 5.5 The environmental impact of RSL changes along the coast of Israel

Another expected effect of the relatively abrupt sea-level rise is the shoreline's landward migration and subsequent coastal morphological change. Unfortunately, in Israel, most of the relevant research so far regarding environmental changes as a response to sea-level rise concern earlier periods than that discussed in the current study. Relatively intensive research however has been carried out In Haifa bay (locally named Zevulun Plain) and around tel Akko in the north of the bay. Zviely et al. [94] found that the sea invasion of Haifa bay started at 8000 to 7150 y BP, reaching about 2 km eastward at 6800 to 6600 y BP. At 4000 y BP (MB age) the coastline was still up to 3 km to the east of the present coastline and from that time onward started a regression trend. Porat et al. [95] confirmed this scenario with Optical Stimulated Luminescence (OSL) ages confirming that halfway between the inner most coastline and the present coast, at 3650 y BP aeolian sand started to accumulate overlying the coastal marine sediments, indicating that the coastline is retreating back rapidly. Elyashive et al. [96] presented fresh wetlands in the back of the coastline, around 7600 to 6200 y BP in the north of the bay and 6500 to 5500 y BP in the south. Only around tel Akko were environmental and human settlement changes observed and dated to the Persian and primarily the Hellenistic periods that are more relevant to the current study. Inbar and Sivan [97] concluded that Middle Bronze to Iron age settlement was located on the calcareous sandstone hill (locally named kurkar) surrounded by brackish water to the south and marine environment to the west and north. In the Persian period the settlement started to move down the hill to the sandy bar that started to accumulate between the tel and the island, now the "old city" of Akko. In the Hellenistic period a sizeable, well-planned city was situated on the plain, west and north-west of the tel. Regarding the harbor of Akko, Morhange et al. [98], followed by Giaime et al. [99] suggest that the MB harbor was located in a marine dominated estuary south of the tel. It moved westward in the Persian period and was finally located in the vicinity of the present-day city of Akko as also claimed by Galili et al. [89] and Galili and Rosen [100]. In Jaffa, Burke et al. [101] published coastal reconstructions but so far without radiometric chronology. About 2000 years ago, Israel's coastal sand unit was thinner by a few meters relative to present thickness [102–104] so it is possible that low laying coastal locations, such as outlets/mouths, were flooded by the post Hellenistic rising sea, creating short live estuaries and lagoons but it seems they were limited in their extent to a few hundred meters. Ongoing geo-archaeological research in a few sites along the coast is expected to add more high-resolution data for these periods.

## 6. Conclusions

- The current study adds a long-term, detailed local record, from a relatively stable coast, to the large number of studies with various sea-level rise rates [103](and references therein). Its

importance goes beyond that of the sea-level research community, as it gives new data that can be used in future archaeological and historical reconstructions.

- The current study adds new, continuous RSL data and re-evaluates previous RSL data for the period of 3800 to 1800 y BP, adding to the existing published data from the last 2000 years [2], and creating a more robust sea-level reconstruction of Israel for the last 3800 years. The most important addition is Iron Age sea levels, a period for which very few previous indicators existed. Various computed model predictions considered in this paper do not agree with the observed relative sea levels, though predictions differ between the models due to different assumptions regarding deglaciation chronology and the Earth's rheological profiles.

- The new RSL index point obtained from the pool in Dor strengthens the conclusion that Roman sea level was near present mean sea level. This new index point is characterized by a high resolution both in elevation measurements and dating, and lies within the range of synthetic RSL predictions obtained with GIA models. The two "end members", the Hellenistic low levels and the Roman high levels, being in close proximity at the same site also imply coastal stability.

- The relatively low sea level between about 3800 y BP and 2200 y BP (MB to Hellenistic period), followed by a rapid rise towards the Roman period, produced a reality whereby, all along the Israeli coast, Hellenistic harbors and anchorages would have been hampered in their activity, and artificial harbor installations would have ceased to serve their function, even if occasionally modified. The economic setbacks and occupational gaps witnessed in the local coastal settlements starting in the 2[nd] century BCE may be connected to this phenomenon, and the nature of this connection should be investigated in greater depth in a separate study. The recovery of some of the impacted settlements and the resumption of harbor construction in the area begin with the relative stabilization of RSL in the Roman period.

## Supporting information

**S1 Table.**
(XLSX)

## Acknowledgments

The Authors gratefully acknowledge the generous support provided by Scripps Center for Marine Archaeology, Scripps Institution of Oceanography, UC San Diego. We thank Ayelet Gilboa, Ilan Sharon and S. Rebecca Martin of the land part of the Tel Dor Joint Expedition for their kind collaboration during the fieldwork. Richard Peltier for distributing online the ICE-X ice sheets chronologies and for providing RSL predictions for some sites of interest. Anthony Purcell is acknowledged for having made the ANU model chronology available to us. All the co-authors also gratefully acknowledge the edition carried out by Dr. S. Dean.

## Author Contributions

**Conceptualization:** Assaf Yasur-Landau, Dorit Sivan.

**Data curation:** Assaf Yasur-Landau.

**Funding acquisition:** Thomas E. Levy.

**Investigation:** Assaf Yasur-Landau, Gilad Shtienberg, Gil Gambash, Ehud Arkin-Shalev, Anthony Tamberino.

**Methodology:** Gilad Shtienberg, Giorgio Spada, Daniele Melini, Jack Reese.

**Resources:** Assaf Yasur-Landau.

**Software:** Giorgio Spada, Daniele Melini.

**Supervision:** Dorit Sivan.

**Visualization:** Gilad Shtienberg.

**Writing – original draft:** Assaf Yasur-Landau, Gilad Shtienberg, Gil Gambash, Dorit Sivan.

**Writing – review & editing:** Assaf Yasur-Landau, Gilad Shtienberg, Gil Gambash, Giorgio Spada, Daniele Melini, Thomas E. Levy, Dorit Sivan.

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
