## [Decision Letter · Decision Letter 0]

15 Apr 2021

PONE-D-21-07251

New relative sea-level (RSL) indications from the Eastern Mediterranean: Middle Bronze Age to the Roman period (~3800-1800 y BP) Archaeological Constructions at Dor, the Carmel coast, Israel.

PLOS ONE

Dear Dr. Yasur-Landau,

Thank you for submitting your manuscript to PLOS ONE. After careful consideration, we feel that it has merit but does not fully meet PLOS ONE’s publication criteria as it currently stands. Therefore, we invite you to submit a revised version of the manuscript that addresses the points raised during the review process.

All comments need to be addressed before re-submission.

We look forward to receiving your revised manuscript.

Kind regards,

Peter F. Biehl, PhD

Academic Editor

PLOS ONE

Journal Requirements:

In your Methods section, please provide additional information regarding the permits you obtained for the work. Please ensure you have included the full name of the authority that approved the field site access and, if no permits were required, a brief statement explaining why.

Please upload a copy of Figure 8, to which you refer in your text on page 17. If the figure is no longer to be included as part of the submission please remove all reference to it within the text.

We note that Figures 1, 2 and 3 in your submission contain map/satellite images which may be copyrighted. All PLOS content is published under the Creative Commons Attribution License (CC BY 4.0), which means that the manuscript, images, and Supporting Information files will be freely available online, and any third party is permitted to access, download, copy, distribute, and use these materials in any way, even commercially, with proper attribution. For these reasons, we cannot publish previously copyrighted maps or satellite images created using proprietary data, such as Google software (Google Maps, Street View, and Earth). For more information, see our copyright guidelines: http://journals.plos.org/plosone/s/licenses-and-copyright.

4a, You may seek permission from the original copyright holder of Figures 1, 2 and 3 to publish the content specifically under the CC BY 4.0 license. 

4b, If you are unable to obtain permission from the original copyright holder to publish these figures under the CC BY 4.0 license or if the copyright holder’s requirements are incompatible with the CC BY 4.0 license, please either i) remove the figure or ii) supply a replacement figure that complies with the CC BY 4.0 license. Please check copyright information on all replacement figures and update the figure caption with source information. If applicable, please specify in the figure caption text when a figure is similar but not identical to the original image and is therefore for illustrative purposes only.

Please include captions for your Supporting Information files at the end of your manuscript, and update any in-text citations to match accordingly. Please see our Supporting Information guidelines for more information: http://journals.plos.org/plosone/s/supporting-information.

Additional Editor Comments:

Your manuscript has now been seen by two referees, whose comments are appended below. You will see from these comments that while the referees find your work of great interest, they have raised some concerns that must be addressed before re-submission.

Reviewers' comments:

Reviewer's Responses to Questions

**Comments to the Author**

1. Is the manuscript technically sound, and do the data support the conclusions?

Reviewer #1: Yes

Reviewer #2: Yes

2. Has the statistical analysis been performed appropriately and rigorously? 

Reviewer #1: Yes

Reviewer #2: Yes

3. Have the authors made all data underlying the findings in their manuscript fully available?

Reviewer #1: Yes

Reviewer #2: Yes

4. Is the manuscript presented in an intelligible fashion and written in standard English?

Reviewer #1: Yes

Reviewer #2: Yes

5. Review Comments to the Author

Reviewer #1: In this paper, the authors aim to establish a more reliable relative sea level curve for the Carmel Coast and Southern Levant between ca. 3500 and 1800 y BP). To establish it, they use a set of new data set acquired recently at Tel Dor. These data are archaeological structures that give both chronological information and relative position according to past sea levels. They clearly reach their mean goal and the new data provided are very valuable. As an overall point of view, the scientific quality of this paper is very hight and very clearly presented, the bibliographical references are up to date and I didn’t notice any lack. The English seems to be very correct, at least for a non-native English speaker as I am.

In the Introduction chapter, the authors present in a synthetic way the general principles guiding the use of archaeological proxies to reconstruct ancient past relative sea level (l. 44-59) and a quick, but precise and complete overview of the state of the research (l. 60-73). They point out a lacuna of information for the period between 4000 and 2000 y BP that they are able to partially fill. The last part of the introduction (l. 74 to 83) presents briefly and superficially the data used and the chronological covering of them. This last is a bit problematic to me. For MBA, LBA and IA, the authors add few words to characterise the periods: “with its second urban revolution” for MBA, “followed by the collapse of the Bronze Age world system” for the LBA and “with its rise of regional states and Assyrian and Babylonian domination” for IA. As a first point, nothing has been added to Persian and Hellenistic period and this disequilibrates the whole passage (l. 77 to 83). The second point is about the interest and the subsequent description of the short characterisation. It is, in my sense, too much or not enough. The quality of the paper does not depend on the characterisation of the period in question and if the authors absolutely want to keep it, it needs to be more detailed and bibliographical references are needed. The way how it is presented is simplistic and disserve the general quality of the paper.

The second chapter precises some regional setting. The authors states first the widely agreed tectonic stability of the Israelian coast (l. 87-102). Numerous and relevant references are provided as well as the main proxies used to demonstrate this stability. A very welcome mention to the very different situation in Lebanon, immediately North of the area in question, is noticeable. Once tectonic stability established, the authors refer to GIA vertical contribution to reconstruct RSL changes. At line 105, the authors mention “Both models present low rate” but only one model is clearly mentioned. The authors state then the importance of the South Levant in term of coastal human occupation and development and how the archaeological sites evolved since Neolithic to Roman period (l. 109-121). Finally, the authors focused on the site of Tel Dor (l. 122-142), which is the main subject of the paper, stating its general chronology and briefly describing the main buildings.

The third chapter describes the methods used to obtain the new data at Dor. Nine new RSL data points, all from Dor, are added to 13 existing on the wider area, and all are presented in the table 1. I do not understand how “new” data points are related to existing bibliography in the table 1. This point needs to be clarified. This chapter is subdivided into 5 subchapters. The first one (l. 161-168), is devoted to underwater structures detailing how the measurements have been done and the dates obtained. All measurements are related to the Israel Land Survey Datum with a zero-mark 8 cm above the mean sea level. It could be interesting to clarify how these 8 cm have been obtained and what the authors called mean sea level (biological sea level?). The second subchapter (l. 161-168) is devoted to the coastal pool at Dor, the authors have created a DSM used to measure the altitude of the sea level indicators and have revisited the unpublished the ceramic assemblage associated to the pool and have corrected the initial datation of the structure by A. Raban (2nd c. CE instead of 2nd-early c. CE). In the third subchapter (l. 171-185) which measurements are relevant to study the past sea levels and why. The fourth subchapter (l. 188-196) deals with uncertainties of the measurements, benchmarks, mean tide fluctuations and how the total uncertainties of the relative sea level estimated has been calculated. The mathematical expression given l. 196 has some formatting errors in my version of the paper that need to be checked. The fifth and last subchapter (l. 199-220) deals with modelling the RSL curve, all the data processing is precisely described including different parameters used (and why they are used) as well as comparative existing models.

The results are presented in the fourth chapter divided into 3 subchapters. The first one (l. 231-285) details the 9 new data set points used in the paper. The identification of the structures used, their dates, the type of RSL information provided and the results of the measurements are listed. The second part (l. 288-309) summarises the new results obtained and initiates a comparison with the already existing data. The authors emphasize already here the main results of this study: (l. 295-298) “the current suggested sea-level position for the period between ~ 3800 y BP and ~2200 y BP was above -3.0 m MSL and probably closer to -2.0 m MSL followed by abrupt sea-level rise to near-present levels in the early Roman period, around 2000-1800 y BP”. In the third subchapter (l. 313-334), the authors compare the model prediction from the data obtained at Dor to different GIA models. No one matches totally with the new data. If most of the differences could be related to “expected range of GIA model uncertainties”, a rapid sea level rise observed for Dor between 2000 and 3000 years “is not reproduced by any of the considered models” (l. 333-334). This very interesting local observation is not discussed in detail in this chapter.

All the data are discussed in the fifth chapter, the last one and the longest one. It is divided into 5 subchapters. In the first subchapter (l. 338-381). The results are put into regional context and allowed the authors to refine the RSL evolution since 3800 to 1800 BP. They focus firstly on a low RSL (between 2 and 3 m below the mean current sea level) dated between 3800 and 2200 BP that could correspond to the altitude of geomorphological features at ca. -2,5 m identified by Goodman-Chernov et al. along the Israeli coast. Secondly, they evaluate the RSL contemporaneous to the activity of the pool at + 0.176m +/- 0.32 that corresponds to the average altitude of the bottom of the channel. Apart the fact that the result only mentioned in chapter 4.1 misses in this part of the paper (only for reading comfort purposes), it is unclear why an average value is used. If we add the maximum uncertainty to this altitude, 0.176 + 0.32= +0.49, sea water was not able to cross a passage at +0.85 m, +0.75 m and +0.66 m before reaching the pool. Something here remains unclear to me. Thirdly, the authors emphasize the “abrupt sea-level rise indicates rates of ~8 mm/year for a few hundred years” (l. 377-378) that is not corroborated elsewhere on the coast without explanation apart the non-tectonic origin of this rise. It could be interesting to suggest any interpretation of it or at least to raise the question. Looking at the fig. 5 (computed relative sea level for the coast of Israel with the chronological and vertical uncertainties), it appears that this rapid rise is mainly based on two data set (n° 15 and 17 in table 1) that do not come from Tel Dor. They come from Akko (n°15) based on Sharvit et al. 2020 and Caesarea (n°17) based on Raban et al. 2009. These two last data sets are not detailed and discussed in the text. Because these forms the angle that mark the beginning of the rapid rise, it seems to me very important to detail them. In the second subchapter (l. 384-434), the authors confront the results of RSL measurements to predicted models. The general conclusion of this part is that no model among the several models tested fit with the data produced in this paper. This have been already said in the subchapter 4.3 with less detail about the characteristics of the models. From my point of view, this part is too long. The figures related are clear enough to avoid the written description of the models tested. Minor changes have to be done in order to avoid repetition between subchapter 4.3 and 5.2. About the discussion itself, I totally understand the authors who spent, without any doubt, many times running the different models. Even if none fit with the new data, it was an important part of this research to test them. As often, absence of result is a result. As mentioned in the paper (l. 431-434) “Various mechanisms to explain fluctuations have been suggested, e.g. [2,19] for the last 2000 years, but so far with no means of verification. The inability, however, of the models to detect the longer-term low sea levels this study presents for the pre-Roman time period requires further investigation”. It could be very welcome to add some further research questions or processes. In the subchapter 3 (l. 437-468)., the authors attempt to put their results in Mediterranean perspective but as they notice (l. 439-441), tectonically stable areas are not common in the Mediterranean Sea. The comparisons used comes from the other part of the Mediterranean, in France and Tunisia and the results, once again, does not fit with the result presented here. I notice here the site of Fréjus (ancient Forum Iulii) is systematically misspell (please change Fre’jus for Fréjus, l. 448, 450, 459…). Despite the fact that no rapid rise is attested in western Mediterranean and low level are lower in Israel than western Mediterranean around ca. 3500 BP, I am asking myself about the relevance of such comparison. I totally understand that the only areas comparable are in western Mediterranean but how to interpret these comparisons? I have no answer apart to put into a light the singularity of the Israeli coastline in the Levant comparing to the neighbouring area (Lebanon and Syria as mentioned l. 464-468). In the fourth subchapter (l. 470-535), the authors raise questions about the impact of the rapid sea level rise observed on coastal structures and sites. They claim, with lot of caution, that this rapid rise could “perhaps”, “among other reasons”, be responsible for the decline of harbour sites and harbour infrastructures precisely listed in the text. The authors point out clearly several examples contemporaneous of this rapid rise and the absence of new harbour structures in the area. They’re right but at the same time the period covering the 2nd and the 1st century BC is not very favourable for flourishing sea-based economy. Meanwhile, it is difficult to agree with such dramatic impact of a, finally, relatively rapid sea level rise compared to tectonically active area, when harbour cities as Tyre, Byzantium or Naples among many others keep important maritime activities despite very important environmental changes or events. As only a personal point of view, the authors are too deterministic. The fifth subchapter (l. 538-565) is devoted to the environmental impact of RSL changes along the Israeli coast. Several examples of environmental from the area are listed with appropriated bibliographical references but none deal clearly with vertical changes of the RSL. All the examples deal with horizontal modification of the coastline (mainly progradation) that is related to a positive sedimentary budget. The relation ship between RSL changes and horizontal changes is possible but not demonstrated.

The conclusions of the paper highlight 4 points. The first one (l. 568-571) states the broad impact of the results presented not only on the “sea-level research community” but as well for “archaeological and historical reconstructions”. It’s clearly the case. The second point (l. 572-578) emphasizes how the new data provided allows to precise the RSL changes especially for a period unwell represented, the Iron Age. The authors mention as well the already detailed differences between the prediction models and their own results stating a fundamental methodological issue. The fourth and the fifth point underline the two major statements of the RSL changes that are (1) a Roman Sea Level next to the current one and (2) the rapid rise of RSL during Hellenistic period from a lower level (ca. 2.5 m) quite stable between 3800 and 2200 BP.

This paper provided new data fundamental for the understanding of the RSL changes along the Israeli coast. As stated by the authors, the results presented will clearly impact future environmental and geoarchaeological research. The data are clear and well presented, the figures are relevant and very readable, the bibliographical references are up to date. My opinion about this paper is very positive but it seems to me that, despite the quality and the importance of the data presented the authors try to go too far within interpretative parts of the text. I am not convinced by the impact on the RSL changes on the coastal sites. The authors clearly and brilliantly state a rapid rise of the RSL during Hellenistic period but didn’t demonstrate the relationship between this rise and the decline of the coastal site. The only argument presented here is chronological parallel. As well, the relationship between environmental changes and RSL changes is not documented. As the authors states (l. 564-565): “Ongoing geo-archaeological research in a few sites along the coast is expected to add more high-resolution data for these periods”.

Finally, a brief format issue to check, in the paper, years are generally abbreviated “y” but I find “yr” as well, e.g. in chapter 4. I recommend the authors to normalise the abbreviations according to the editor recommendations.

Reviewer #2: This is my first review where I do not have any substantive recommendations to the authors - the manuscript is in publishable form in my opinion. All the concerns about the data I would have (subsidence, effects from other Mediterranean sites) are addressed by the authors in a convincing manner. I find it surprising that a substantive sea level change is associated with a comparatively stable period in the history of the region.

6. PLOS authors have the option to publish the peer review history of their article (what does this mean?). If published, this will include your full peer review and any attached files.

Reviewer #1: No

Reviewer #2: No

---

## [Author Response · Author response to Decision Letter 0]

29 Apr 2021

Cover letter and respond to reviewers following the review for:

New relative sea-level (RSL) indications from the Eastern Mediterranean: Middle Bronze Age to the Roman period (~3800-1800 y BP) Archaeological Constructions at Dor, the Carmel coast, Israel.

Dear Prof. Peter F. Biehl and PLOS ONE editorial board,

We wish to thank you and the anonymous referees for the time and effort devoted to reviewing and improving our manuscript. We have accepted the majority of comments and suggestions and followed your instructions carefully while preparing the response document (please see the response to reviewers document). 

Below, we present each editorial comment, followed by the corrected text and /or specific responses. In order to distinguish between them, the editorial comments appear in red font, and our responses and edited texts are typed in blue. 

We hope you will find this version suitable for publication in PLOS ONE. 

Sincerely,

Prof. Assaf Yasur-Landau –

Department of Maritime Civilizations,

The Leon H. Charney School of Marine Sciences,

The Leon Recanati Institute for Maritime Studies (RIMS)

University of Haifa, Haifa 3498838, Israel

Editorial comments

1. Please ensure that your manuscript meets PLOS ONE's style requirements, including those for file naming

The required modifications have been made.

The permit information was added to the Methods section.

3. Please upload a copy of Figure 8, to which you refer in your text on page 17. If the figure is no longer to be included as part of the submission please remove all reference to it within the text.

The manuscript consists of a total of 7 figures. The reference to figure 8 was erased from the text.

4. We note that Figures 1, 2 and 3 in your submission contain map/satellite images which may be copyrighted.

Appropriate copyrights were acquired from the Geological Survey of Israel (please see correspondence below) for a 25 m raster grid that captures the surface elevation of the eastern Mediterranean. This file was the basis for the creation of figure 1b. Figures 2 and 3 were created by data that was generated as part of the current research.

The GSI confirmation: 

Dear Gilad, 

We are happy to provide explicit permission for use of data from the Geological Survey of Israel that was made in the figure attached under the CC BY 4.0 license. 

Please verify adequate citation of John Hall’s data in your manuscript. 

All the best,

Dr. Amit Mushkin

-----

Head, Geological Mapping Division

Geological Survey of Israel, 

Jerusalem, Israel

Figure 1a (Israel in the SE Mediterranean) was created through modification of (https://www. naturalearthdata.com in the public domain) and this information was added to the figure caption while 1c (aerial photograph) and 3 were created in the current study using a drone and implementing photogrammetry technics. 

The required modifications have been made.

The required modifications have been made.

Reviewer 1 comments

1. The last part of the introduction (l. 74 to 83) presents briefly and superficially the data used and the chronological covering of them. This last is a bit problematic to me. For MBA, LBA and IA, the authors add few words to characterise the periods: “with its second urban revolution” for MBA, “followed by the collapse of the Bronze Age world system” for the LBA and “with its rise of regional states and Assyrian and Babylonian domination” for IA. As a first point, nothing has been added to Persian and Hellenistic period and this disequilibrates the whole passage (l. 77 to 83). The second point is about the interest and the subsequent description of the short characterisation. It is, in my sense, too much or not enough.

In agreement with the reviewer’s suggestion, the characterizations of the MBA, LBA and IA were erased from the text.

2. At line 105, the authors mention “Both models present low rate” but only one model is clearly mentioned.

The sentence was corrected as follows:

The new paragraph now is: “As for Glacial Isostasy Adjustment (GIA) vertical contribution, the models produced for Israel by Lambeck [1,11,36] indicates rising RSL throughout the Holocene. Unlike these models, for the last 4000 years, ICE-5G [19,37,38] predicts RSL falling to present levels. In all cases, these low rates of RSL change are attributed to GIA, which varies between < 0.2 mm/y for the last 8000 years [1] and 0.15 mm/y in the last 1000 years [19], amounting to ~20 cm RSL rise in 1000 years. For the last 2000 years, the reconstructed RSL that is based on observations indicates small fluctuations above and below present mean sea level as summarized in Dean et al. [2].”

3. The third chapter describes the methods used to obtain the new data at Dor. Nine new RSL data points, all from Dor, are added to 13 existing on the wider area, and all are presented in the table 1. I do not understand how “new” data points are related to existing bibliography in the table 1. The point needs to be clerefide

As suggested, this point was clarified in section 3:

“The current research adds 9 new RSL data points (all those from Dor) to the 13 existing points (Fig. 2 and Table 1), thus significantly improving the resolution of sea-level change in the Southern Levant between the Middle Bronze Age and the Roman period. The new RSL points presented below were established by transforming recently excavated archeological constructions elevations and various functions into RSL data. This information was not used so far as sea level indicators.”

4. All measurements are related to the Israel Land Survey Datum with a zero-mark 8 cm above the mean sea level. It could be interesting to clarify how these 8 cm have been obtained and what the authors called mean sea level.

Following this comment, we have revised section 3.1 regarding the Israeli land survey datum (ILSD) - MSL transformation which now reads as follows:

“Measurement and levels were taken using a Leica TS9 Plus Total Station as well as a Leica TS06 Plus with an error of ±5 and ±10 cm respectively relative to Israel Land Survey Datum (ILSD). Subsequently, the ILSD points were converted into local MSL, following Rosen et al. [66] calculations. Based on these finds, that relied on tidal gauges measurements distributed along the Israeli coast between 1958 – 1984, it was determined that MSL is higher by 8 cm above ILSD due to ongoing sea level rise.”

Still related to this comment, we have rechecked the transformation calculation between the Israeli land survey datum (ILSD) to MSL in the original excel spread sheet and found a calculation error. This error consisted of subtraction of 8 cm from the measured elevation relevant to ILSD instead of adding 8 cm to the measured elevation when converting to MSL. This issue was corrected in the excel spread sheet, in table 1 as well as in figure 5.

We have also added an asterisk to the original excel spread sheet column representing an indicator that was measured compared to the Israeli land survey datum and which was then converted into MSL.

5. The mathematical expression given l. 196 has some formatting errors in my version of the paper that need to be checked.

The mathematical expression was reformatted.

6. If most of the differences could be related to “expected range of GIA model uncertainties”, a rapid sea level rise observed for Dor between 2000 and 3000 years “is not reproduced by any of the considered models” (l. 333-334). This very interesting local observation is not discussed in detail in this chapter.

We feel that the observation is sufficiently discussed in section 5.2, maintaining balance with the other parts of the article.

7. Apart the fact that the result only mentioned in chapter 4.1 misses in this part of the paper (only for reading comfort purposes), it is unclear why an average value is used. If we add the maximum uncertainty to this altitude, 0.176 + 0.32= +0.49, sea water was not able to cross a passage at +0.85 m, +0.75 m and +0.66 m before reaching the pool. Something here remains unclear to me.

In accordance with this comment, we revised section 3.3b as follows:

“Two assumptions serve in the background of employing the rock-cut pool as a tool for the reconstruction of past RSL: a. when the pool was in service, it would have received seawater through wave activity higher than 0.40 m (which is common for the coast of Israel); and b. the RSL could not have been higher than the elevation of the pool's rim, or lower than its base [7,8,68]. Thus, in Dor, the channel feeding the pool with seawater would have functioned between these limits. Elevations of the channel's base rise gradually from +0.1m at the channel's connection with the sea, to +0.75m at the channel's connection with the pool - probably in order to reduce wave-energy and encourage the settling of sand prior to the water's entry to the pool (Fig. 3). Therefore, we relied on 30 elevation points measured along the axial center of the first 15 meters of the seaward part of the channel. An average was then calculated for these elevation points, which enabled us to reduce the elevation influence of recent bio-rocks formed on the host sandstone aeolianite platform and other post-usage influences.”

8. the authors emphasize the “abrupt sea-level rise indicates rates of ~8 mm/year for a few hundred years” (l. 377-378) that is not corroborated elsewhere on the coast without explanation apart the non-tectonic origin of this rise. It could be interesting to suggest any interpretation of it or at least to raise the question.

Following this comment, we have erased the calculated rate. The revision reads as follows:

 “The abrupt seal-level rise which started during the Hellenistic period along the coast of Israel came to end at a certain point during the Roman period, and since that point, for roughly 2,000 years, the RSL has remained the same, with the exception of several short-term fluctuations, as shown by Toker et al. [19] and Dean et al. [2]. These new finds and past records presented here cannot be explained so far by a known mechanism.”

9. Looking at the fig. 5 (computed relative sea level for the coast of Israel with the chronological and vertical uncertainties), it appears that this rapid rise is mainly based on two data set (n° 15 and 17 in table 1) that do not come from Tel Dor. They come from Akko (n°15) based on Sharvit et al. 2020 and Caesarea (n°17) based on Raban et al. 2009. These two last data sets are not detailed and discussed in the text. Because these forms the angle that mark the beginning of the rapid rise, it seems to me very important to detail them.

One of the contributions of the current paper is the new, higher resolution RSL data (both elevations and dating) obtained for the Hellenistic period in Akko and for the Roman period in the pool of Dor. Detailed measurements of the Akko anchorage are now available in the new publication:

Sharvit, J. Buxton, B. Hale, J.R. Ratzlaff A, The Hellenistic-Early Roman Harbour of Akko 2021. In Demesticha, S. and Blue, L. et al. (eds), Under the Mediterranean I. Studies in Maritime Archaeology, Leiden: Sidestone Press, pp. 163-180.

The reference for this new publication was added throughout the revised version of the manuscript. 

10. These two last data sets (Akko and Caesarea) are not detailed and discussed in the text. Because these forms the angle that mark the beginning of the rapid rise, it seems to me very important to detail them.

We agree with the comment and have added accordingly the following sentence to section 5.1.

“These low levels existed up to ~2200 BP (Hellenistic period), then RSL began to rise. This sea level change is best confirmed by RSL difference between the recently published Hellenistic harbor installations [48] in Akko, dated to 2175 BP, and the pools of Dor, dated to the Roman period. While the Akko port installations are at an elevation of 1.1 m below MSL [see Table 1; Fig. 4], the pools of Dor are at near-present RSL [2,11,26]”.

11. The general conclusion of this part is that no model among the several models tested fit with the data produced in this paper. This have been already said in the subchapter 4.3 with less detail about the characteristics of the models. From my point of view, this part is too long.

We agree with the comment and have shortened this part by removing the description of the characteristics of the models, referring the reader to the discussion in section 4.3.

12. I notice here the site of Fre’jus (ancient Forum Iulii) is systematically misspell (please change Fre’jus for Fre’jus, l. 448, 450, 459…).

As suggested, the text was corrected.

13. I totally understand that the only areas comparable are in western Mediterranean but how to interpret these comparisons? I have no answer apart to put into a light the singularity of the Israeli coastline in the Levant comparing to the neighboring area (Lebanon and Syria as mentioned l. 464-468).

Unfortunately, due to its tectonic stability and relatively low isostatic nature, only certain parts of the western basin of the Mediterranean could be relevant for such comparison. However, such areas also necessitate a continuous record of coastal habitation that resemble those of the Israeli coast and the uncertainties both in the western sites and in the Israeli record are still relatively large. 

14. Meanwhile, it is difficult to agree with such dramatic impact of a, finally, relatively rapid sea level rise compared to tectonically active area, when harbour cities as Tyre, Byzantium or Naples among many others keep important maritime activities despite very important environmental changes or events. As only a personal point of view, the authors are too deterministic.

I am not convinced by the impact on the RSL changes on the coastal sites. The authors clearly and brilliantly state a rapid rise of the RSL during Hellenistic period but didn’t demonstrate the relationship between this rise and the decline of the coastal site. The only argument presented here is chronological parallel. As well, the relationship between environmental changes and RSL changes is not documented. As the authors states (l. 564-565): “Ongoing geo-archaeological research in a few sites along the coast is expected to add more high-resolution data for these periods”.

We agree that a more balanced approach should be adopted, making the rise in RSL part of a larger set of circumstances affecting resilience along the coastal southern Levant during the Hellenistic period. Modifications have been made to key statements accordingly, and particularly to the analysis in section 5.4, now reading:

“The more direct impact of the rapid rise of RSL would have been felt in the sphere of marine installations, and the economy which closely relied on them. Settlements which employed built coastal features would have been vulnerable as far as their maritime related endeavors were concerned, since the installations on which they relied would have gone out of use at a rapid pace, even if they attempted to repair and adjust them to the rising sea level. Both the means to protect harboring ships, such as the sea-walls of Akko-Ptolemais [48], and the installations devised to facilitate the loading and unloading of cargo, such as the quays of Dor and Straton’s Tower, would have been rendered dysfunctional as a result of the significant rise in RSL. Larger cities with more varied economies, such as Akko-Ptolemais and Dor, would have been more resilient, and continued to exist during the challenging period, though their maritime-based activity would have been reduced. But a long list of smaller coastal settlements show clear signs of collapse, including Yavneh Yam, Ashdod Yam, Straton’s Tower, and tel Taninim.”

15. All the examples deal with horizontal modification of the coastline (mainly progradation) that is related to a positive sedimentary budget. The relationship between RSL changes and horizontal changes is possible but not demonstrated.

We appreciate the offer to expand the section discussing horizonal modification of the coastline. However, we are concerned that, in order to do justice to the topic, we would have to include a plethora of data worthy of a new paper. Accordingly, we prefer to leave the in-depth sea level transgression and coastal evolution discussion for the future.

16. Finally, a brief format issue to check, in the paper, years are generally abbreviated “y” but I find “yr” as well, e.g. in chapter 4. I recommend the authors to normalise the abbreviations according to the editor recommendations.

As suggested, the text was corrected.

---

## [Editor Report · Decision Letter 1]

5 May 2021

New relative sea-level (RSL) indications from the Eastern Mediterranean: Middle Bronze Age to the Roman period (~3800-1800 y BP) Archaeological Constructions at Dor, the Carmel coast, Israel.

PONE-D-21-07251R1

Dear Dr. Yasur-Landau,

We’re pleased to inform you that your manuscript has been judged scientifically suitable for publication and will be formally accepted for publication once it meets all outstanding technical requirements.

Kind regards,

Peter F. Biehl, PhD

Academic Editor

PLOS ONE
---

## [Editor Report · Acceptance letter]

12 May 2021

PONE-D-21-07251R1 

New relative sea-level (RSL) indications from the Eastern Mediterranean: Middle Bronze Age to the Roman period (~3800-1800 y BP) Archaeological Constructions at Dor, the Carmel coast, Israel. 

Dear Dr. Yasur-Landau:

I'm pleased to inform you that your manuscript has been deemed suitable for publication in PLOS ONE. Congratulations! Your manuscript is now with our production department. 

Kind regards, 

on behalf of

Dr. Peter F. Biehl 

Academic Editor

PLOS ONE